# Mitochondria at Work: New Insights into Regulation and Dysregulation of Cellular Energy Supply and Metabolism

**DOI:** 10.3390/biomedicines8110526

**Published:** 2020-11-22

**Authors:** Volker Schirrmacher

**Affiliations:** Immune-Oncological Center Cologne (IOZK), D-50674 Cologne, Germany; V.Schirrmacher@web.de

**Keywords:** chronic diseases, OXPHOS, TCA, cyanobacteria, glycosylation, cancer, tumor microenvironment, cachexia, hydrogen, oxygen, redox enzymes

## Abstract

Mitochondria are of great relevance to health, and their dysregulation is associated with major chronic diseases. Research on mitochondria—156 brand new publications from 2019 and 2020—have contributed to this review. Mitochondria have been fundamental for the evolution of complex organisms. As important and semi-autonomous organelles in cells, they can adapt their function to the needs of the respective organ. They can program their function to energy supply (e.g., to keep heart muscle cells going, life-long) or to metabolism (e.g., to support hepatocytes and liver function). The capacity of mitochondria to re-program between different options is important for all cell types that are capable of changing between a resting state and cell proliferation, such as stem cells and immune cells. Major chronic diseases are characterized by mitochondrial dysregulation. This will be exemplified by cardiovascular diseases, metabolic syndrome, neurodegenerative diseases, immune system disorders, and cancer. New strategies for intervention in chronic diseases will be presented. The tumor microenvironment can be considered a battlefield between cancer and immune defense, competing for energy supply and metabolism. Cancer cachexia is considered as a final stage of cancer progression. Nevertheless, the review will present an example of complete remission of cachexia via immune cell transfer. These findings should encourage studies along the lines of mitochondria, energy supply, and metabolism.

## 1. Introduction

Energy and the environment are important and timely issues in present-day discussions. Engineers develop techniques to use hydrogen (H) instead of carbon (C) as fuel, and sunlight (photons) as solar energy to burn H_2_, with the help of oxygen (O_2_), into water molecules. Nature invented its own technology 1.5 to 3.5 billion years ago. It uses H_2_, O_2_, and CO_2_ to generate glucose for metabolism and energy production. The ins-and-outs of the gases O_2_ and CO_2_ are in balance with the environment. Details about the evolution of life on earth can be found in an excellent textbook [1].

Cyanobacteria invented photosynthesis, a process to assimilate C from CO_2_ for sugar (glucose) synthesis. The energy was taken from photons and electrons (e^−^). In those times, without O_2_ in the atmosphere, nicotinamide adenine dinucleotide (NAD) was used as the redox system, NAD/NADH. It served to remove H from H_2_O molecules and generate protons (H^+^), electrons (e^−^), and the reduction equivalent NADH, according to the formula: NAD^+^ + 2H = NADH + H^+^ + e^−^. Archaebacteria used a process of anaerobic glycolysis to degrade glucose and to produce and store energy in the form of adenosine triphosphate (ATP). Since those times, ATP has served as a main energy source in cells on earth. Once the atmosphere was enriched with O_2_ (>1%), eubacteria invented aerobic glycolysis of sugars via oxidative phosphorylation (OXPHOS), resulting in more efficient production of ATP. During further evolution towards eukaryotic cells, nature developed membrane-enclosed nuclei and cellular organelles specialized for energy supply and metabolic processes: mitochondria and chloroplasts. These organelles are equipped with biomembrane-associated or matrix-embedded enzymes to catalyze biochemical redox processes, involving electron transport chains and metabolic processes. Magnesium, containing chlorophyll in chloroplasts, absorbs and transfers energy from photons. Iron, containing cytochromes in mitochondria, with hem as cofactor, transport electrons to generate ATP. The iron of hemoglobin reacts in a reversible manner with molecular oxygen, in which one atom of iron combines with one molecule of oxygen [2]. Multiple lines of recent evidence support the hypothesis that mitochondria and chloroplasts were transferred from bacteria to eukaryotic cells, most likely via endocytosis and symbiosis. The cytoplasm of eukaryotic cells, in contrast to these organelles, exerts anaerobic glycolysis, such as in ancient times. Mitochondria have been (and are) fundamental for the emergence of metazoans [3]. The perpetual work of these cellular organelles over billions of years under changing environmental conditions is of great fascination.

Aerobic and “truncated” aerobic glycolysis are examples of regulated or dysregulated biochemical processes in various types of cells [2]. The role of mitochondria in these processes will be described and discussed. Heart muscle cells will be used as an example of cells degrading pyruvate via OXPHOS, to its full extent, for maximal ATP production. Skeletal muscle cells serve as an example for cells, which, under stress, degrade pyruvate via fermentation to lactate. Liver cells serve as an example of cells busy with mitochondrial metabolic (anabolic and catabolic) processes (e.g., glycogen storage, detoxification).

New insights will be provided into the metabolism of cells from the immune system, in particular T lymphocytes and their various subsets. Upon activation, T cells switch their metabolism from a resting state to a state allowing cell proliferation and defined immune functions. Distinct transcription factors will be mentioned, which have been identified as key regulators of metabolism of T cell subsets, such as naive, Th1, Th2, Th9, Th17, Treg, and memory T cells (MTCs) [4].

Mitochondria also play important roles in pathophysiological processes of diseases of major importance, such as cardiovascular disease (CVD) and cancer. Recent research has identified a variety of features of metabolic dysregulation, such as “truncated” aerobic glycolysis in cancer that could become targets of new therapeutics. The Warburg effect of aerobic glycolysis [5] fulfills the needs, not only of cancer cells, but also of an activated immune system. This appears of particular significance in the tumor microenvironment (TME) where cancer cells and immune cells compete for supply of energy and nutrients.

Of particular significance are recent findings related to cancer cachexia. Little is known, so far, in molecular terms, about this muscle and fat waste syndrome. Meanwhile, mitochondrion dysfunction, new receptor signaling pathways, transcription factors, regulatory RNA, hindbrain and adipose tissue hormones, and immune system factors have been identified. This review will provide one example of reversion of cancer cachexia in a Graft-versus-Leukemia (GvL) mouse model system via the transfer of specific immune T lymphocytes [6]. Such results from basic research should encourage further studies on immunotherapy to interfere with chronic diseases.

## 2. Structure, Function, and Potential Origin of Mitochondria

The first observation of a metabolic alteration in cancer cells was made about a century ago [2,5]. This Warburg effect of aerobic glycolysis, with excess lactate production, became, only recently, an active research field, termed metabolic reprogramming. Metabolism refers to all biochemical processes of a cell, and of a multicellular organism, to maintain steady-state homeostasis and survival. What will be summarized first is a prerequisite to understand the newest discoveries of mitochondrial research. The basis is described in biochemistry, cell biology, and physiology textbooks.

Phosphate metabolism, for example, includes, among others, biochemical energy transfer via ATP, maintenance of genetic information with DNA and RNA nucleotides and membrane structural integrity via glycerophospholipids [7].

Table 1 presents an overview of energy production by bacteria. During anaerobic times, the energy source for electrons was, for example, sulfur (H_2_S), oxygen (H_2_O), or methane (CH_4_). As prokaryotic cells, archaebacteria (methanogenic, sulfurogenic, and thermoacidophilic) started to sequester their genome, which later led to development of a membrane-enclosed nucleus. Archaebacteria and early eubacteria had already developed the processes of glycolysis (catabolism of glucose to pyruvate) and fermentation (e.g., catabolism of pyruvate to lactate), with all their necessary enzymes and co-enzymes. These types of anaerobic metabolism have survived times, and are still at work in the cytoplasm of plant and animal cells, including Homo sapiens. Important coenzymes are NAD^+^ and its phosphate (NADP^+^), coenzyme A (CoA), and flavin adenine dinucleotide (FAD). Glycolysis is a term for degradation of glucose to pyruvate with the help of adenosine diphosphate (ADP) and inorganic phosphate (Pi). It involves 10 enzymatic steps and works according to the formula: 1 glucose + 2 ADP + 2 Pi + 2 NAD = 2 pyruvate + 2 ATP + 2 NADH2 [1].

OXPHOS in mitochondria starts from pyruvate, which enters the mitochondrial matrix (MM) to become decarboxylated and converted into acetyl-CoA. This 2-carbon molecule then condenses with a 4-carbon molecule to produce the 6-carbon molecule citrate. The core of cellular metabolism is found in the citrate or tricarboxylic acid cycle (TCA or citrate cycle). Citrate is decarboxylated successively to generate 5-carbon and 4-carbon molecules. The released CO_2_ diffuses to the outside. After conversion to the 4-carbon molecule oxaloacetate, the TCA cycle can start again being driven by acetyl-CoA. The four NADH molecules that are generated per cycle diffuse to the inner mitochondrial membrane (IMM) and transfer their hydrogen reduction equivalents into the electron transport chain (ETC) for generation of energy. This ETC respiration chain, catalyzed by iron (Fe^3+^ + e^−^ = Fe^2+^), serves to transfer electrons from a high level of energy to a level of low energy. It is made up of five enzyme complexes: NADH dehydrogenase (complex I), succinate dehydrogenase (complex II), cytochrome-c reductase (complex IV), cytochrome-c oxidase (complex IV), and ATP synthase (complex V). An electrochemical proton gradient (ΔH^+^) between the peri-mitochondrial cleft and the MM drives the production of ATP from ADP and Pi, catalyzed by the ATP synthase. The energy of this gradient also drives specific inner mitochondrial membrane (IMM) transporters for Pi, pyruvate and Ca^2+^. The mitochondrial production of ATP is done by oxidizing the major products of glycolysis from the cytosol: pyruvate and NADH.

The TCA cycle involves five enzymatic steps to generate energy and to convert small metabolites into precursors for biosynthetic pathways. It finally produces from protons, electrons, and oxygen water, according to the formula: 4H^+^ + 4e^−^ + O_2_ = 2H_2_O. This burning of hydrogen is facilitated by cytochrome enzymes in the IMM. Mitochondrial OXPHOS and ETC respiration produces approximately 13-times more ATP per molecule glucose than glycolysis. The ATP produced in mitochondria crosses out through the IMM, with the help of a specific protein carrier, and across the outer mitochondrial membrane (OMM) via porins into the cytoplasm. ADP returns via the same route.

Apart from supplying energy, mitochondria are involved in other cellular processes: signaling, differentiation, maintenance of cell cycle and cell growth, and cell death (intrinsic apoptosis following cytochrome-c release). They are also involved in catabolic and anabolic biochemical processes, often upon interaction with the cell’s cytoplasm. Examples of such catabolic processes are ketone body degradation (e.g., via acetyl-CoA-acetyl-transferase) and the urea cycle for elimination of NH_3_. Examples for anabolic processes are gluconeogenesis and the pentose phosphate pathway, leading to de novo purine biosynthesis. Such metabolites are used for synthesis of macromolecules, such as hem and porphyrins, steroid hormones, RNA, and DNA.

All of this textbook knowledge is important to understand how mitochondria work in different cell types under physiological or pathophysiological conditions. This has consequences for diagnosis and therapy of mitochondria-associated symptoms [8].

Interestingly, cyanobacteria and plant chloroplasts perform a biochemical reaction, which is in the opposite direction compared to mitochondria. They photolyse water (2H_2_O = 4H^+^ + 4e^−^ + O_2_) with the help of sun light to generate energy. For comparison: mankind tries to generate energy from water by electrolysis. Nature invented for enzymatic catalysis of this reaction intracytoplasmic sophisticated thylakoid membrane systems containing pigments, such as chlorophyll, carotenoids, and phycoerythrin. These are able to absorb light, of wavelengths 560 to 670 nm. In the light reaction, chloroplasts transfer energy to electrons at a high orbital and produce NADPH and ATP. This allows, in the dark reaction, to assimilate CO_2_ to produce glucose (Calvin cycle). The photosystem I, with its ferredoxin-NADP^+^-reductase produces NADPH + H^+^, and the photosystem II, with its ATP synthase (dark reaction) produces, finally, the energy “currency” ATP, in which phosphate metabolism plays an important role [7]. Conservation and conversion of energy in plant cells involves (1) chloroplasts (generation of NADPH, ATP, glucose, and starch); (2) cytosol (degradation of glucose or starch to pyruvate involving glyceraldehyde-3-phosphate dehydrogenase (GAPDH); and (3) mitochondria (degradation of pyruvate via TCA and ETC to ATP).

Without the synthesis of glucose and starch by plant cells, the energy metabolism of animal cells would not be possible. Plant products, serving as nutrients, become degraded in the intestine before their metabolites (e.g., glucose, fatty acids) are permitted to enter the blood stream. Chloroplasts take up three molecules of CO_2_ and release three molecules of O_2_ per molecule glucose. Mitochondria take up three molecules of O_2_ and release three molecules of CO_2_ per molecule glucose. There is, thus, a net balance between consumption and waste production executed, and finely tuned by the cellular organelles mitochondria and chloroplasts.

Of particular importance in both organelles are the electron transport chains mediated by redox enzymes. Models of quantum theory locate electrons on orbitals surrounding the atom at defined distances. The higher the main quantum number, the further away the electron and the more reactive it can be in redox systems. Electrons at δ-orbitals of enzyme A, with high electron density, transfer electrons to enzyme B with lower electron density. In this way, the reducing enzyme A becomes oxidized and the oxidized enzyme form B becomes reduced. Many biological redox systems contain ions from metals (e.g., Fe, Cu, and Mn); others contain disulfide bridges (e.g., glutathione, thioredoxin). Important coenzymes are based on quinone (e.g., ubiquinone (Q10), plastoquinone, vitamin E, and K), nicotinamide (e.g., NAD, NADP), or flavin (e.g., FAD). Q10 is a vitamin-like compound. It is expressed in humans as ubiquinol (reduced form) or ubiquinone (oxidized form). It plays a key role in electron transport in OXPHOS. It acts as a potent antioxidant, membrane stabilizer, and cofactor in the production of ATP.

Table 2 provides an overview of the structure of mitochondria and compares them with chloroplasts. Both eukaryotic organelles have the size of bacteria and are surrounded within the eukaryotic cell by a double membrane. The IMM consists of lipids characteristic for bacteria (e.g., cardiolipin (CL)), while the outer membrane consists of lipids and porin molecules characteristic for eubacteria. Between the two membranes exists a space, which is used to establish the above-mentioned proton gradient for the ATP synthase pump.

The OMM contains integral membrane proteins called porins. One of these trafficking molecules is the voltage-dependent anion channel (VDAC), a transporter of nucleotides, ions, and metabolites between the cytosol and the intermembrane space. The inner membrane contains proteins of the electron transport chain redox reactions, ATP synthase, and specific transport proteins that regulate metabolite passage into and out of the mitochondrial matrix. Depending on the mechanism and direction of transported molecules, one differentiates between antiporter, symporter, and uniporter. The driving force comes either from the membrane potential (e.g., Ca^2+^ influx and ATP/ADP exchange) or from proton gradients (e.g., import of pyruvate and Pi).

The inner membrane is compartmentalized into cristae, which expand the surface area. For typical liver mitochondria, the area of the inner membrane is about five times the size of the outer membrane. Cells that have a high demand for ATP, such as muscle cells, contain even more cristae. The IMM contains more than 151 different polypeptides. One of these is the phospholipid CL, which is coded by mitochondrial DNA (mtDNA). It contains four fatty acids rather than two. These many fatty acids apparently help to make the inner membrane impermeable.

The matrix, the space enclosed by the IMM, is the site where ATP is being produced. It contains a concentrated mixture of enzymes, special mitochondrial ribosomes, tRNA, and several copies of the mitochondrial DNA genome. The enzymes facilitate oxidation of pyruvate and fatty acids, and running of the TCA cycle. The reduction equivalents NADH and FADH2 are produced within the matrix via the TCA cycle but are also produced in the cytoplasm by glycolysis. The electrons from NADH and FADH2 are transferred to oxygen (O_2_), an energy-rich molecule, and hydrogen (protons) in several steps via the ETC.

Mitochondria have their own genetic material in the form of circular DNA. Moreover, they have the machinery to produce their own RNAs and proteins. A human mitochondrial DNA sequence revealed 16,569 base pairs encoding 37 genes: 22 tRNAs, 2 rRNAs, and 13 peptide genes [8]. Like in bacteria, the DNA has a ring structure without histone and one start site of replication. The start of protein synthesis via bacteria-like ribosomes occurs, such as in bacteria, via N-formyl-methionine. Human mitochondrial DNA is double-stranded and is passed from the mother to her offspring during reproduction. Such DNA probes are being used in forensic medicine and in paleo-anthropology. Similarities between human mitochondria and bacteria have recently been described also with respect to the DNA base excision repair (BER) system [9].

Mitochondrial nuclear genes POLG, POLG2, TWNK, and SSBP1 encode the core mtDNA replisome. The mtDNA is replicated by a core set of proteins: polymerase γ, Twinkle, and the single-stranded DNA binding protein [10]. Polymerase γ efficiently replicates through many natural template barriers (e.g., double stranded DNA, structured genes, G-quadruplexes) [11].

The molecular machineries for mitochondrial fusion and fission are essential for mitochondrial homeostasis in health and disease. The fusion of the OMM and the IMM is mediated by dynamin-like proteins (DLPs) [12]. Mitochondrial fission is likely due, among others, to increasing FIS1 and decreasing Mfn2 [13]. An example of the consequence of too much fission is cigarette smoke-induced pulmonary endothelial injury [13]. An example of the consequence of too much fusion is COVID-19. Based on new and compelling evidence, it is proposed that fusion is promoted, causing mitochondrial elongation, and providing a receptive intracellular environment for viral replication in infected cells [14].

The hypothesis of an endosymbiotic relationship of mitochondria with their host cells was popularized by Lynn Margulis in 1986 [15]. An alternative hypothesis, the autogenous hypothesis, is nowadays less widely accepted. The symbiotic relationship was probably developed 1.7 to 2 billion years ago. It appears likely that mitochondria developed earlier than chloroplasts. This might explain that the copy number of ptDNA is about tenfold higher than that of mtDNA (Table 1). It might also explain that animal and plant cells have mitochondria, and that only plant cells have, in addition, chloroplasts. The reduction equivalents, NADH of animal cells and NADPH of plant cells, contain high-energy electrons, and are, thus, being used as fuel to produce ATP.

In terms of evolution, the incorporation of mitochondria into the eukaryotic cell facilitated the use of oxygen (for OXPHOS and respiration) and its detoxification (to reduce oxidative stress induced by reactive-oxygen-species (ROS)). The higher the mitochondrial metabolism, the higher the production of ROS. For protection against their toxic effects, mitochondria developed enzymes, such as superoxide dismutase (SOD), catalase, glutathione peroxidase (GSH-Px), glutathione transferase, and cytochrome-c-peroxidase. These enzymes neutralize ROS by finally producing water molecules [16].

Symbiosis between bacterial organelles and eukaryotic cells over time of evolution has likely optimized the interaction between the two types of worlds. Import of metabolites from the cell’s cytoplasm to the organelles occurs freely or via porin molecules. Export from the organelles to the cytoplasm, however, does not occur. Some of the inner membrane proteins are only partially encoded by organelle-specific DNA. Other parts of these enzyme complexes are derived from the eukaryotic cell. The majority of mitochondrial proteins are cell-nucleus encoded. Mitochondria, the powerhouse of the cell, are semi-autonomous cell organelles.

These insights into the origin (evolution), structure (molecular and cell biology), and function (biochemistry, physiology) of mitochondria are considered important to understand the pathophysiology of major chronic human diseases, such as CVD, metabolic syndrome (MetS), neurodegenerative diseases, immune system disorders, and cancer.

## 3. Biological Hydrogen Production: An Alternative Energy Source

As stated at the beginning, energy and environment are important issues of present day discussions. Since this is not the main issue of this review, only a few aspects can be addressed here.

Hydrogen is the first and most abundant element of the universe. From the fusion of hydrogen atoms, stars receive their energy. The energy from our sun keeps our planet warm and provides electromagnetic waves, including visible light (photons). This is used by pigments in chloroplasts of plants for photosynthesis and by pigments in mitochondria of animal retinal photoreceptor cells for eye vision. Hydrogen is a clean fuel having a high-energy content (122 kJ/g). It can be used for electricity generation by fuel cells or in combustion engines. It is also an energy source of interest with regard to environmental protection.

Recent trends in hydrogen energy have been discussed [17]. Fermentative hydrogen production experiments from food-based industrial waste have identified four effective bacterial strains. Klebsiella pneumoniae-FA2 surpassed the other three with a maximum hydrogen yield of 3.79 mol H_2_/mol glucose [18]. Bioelectrical systems are being studied as sustainable bio-energy powerhouses [19]. Microbial electrosynthesis systems are being evaluated to produce multicarbon chemicals from CO_2_ [20]. The major problems in large-scale biological hydrogen production are the low production rate and yield. Therefore, mixed culture technologies are being recommended [21]. Algae-bacteria consortia are being studied as model to improve biohydrogen production [22]. Since mitochondria are very likely derived from bacteria, it would be very meaningful to use bacteria as bio-energy powerhouse.

We now go to mitochondria at work in our cells.

## 4. ATP Supply for the Muscle

### 4.1. Physiologic Situation

Regulation of tissue oxygenation [23] includes the respiratory system, the blood circulatory system, and the cardiovascular respiratory system. Oxygen that has diffused into cells is utilized in mitochondria to produce ATP. Mitochondria are able to produce ATP until the oxygen tension or P_02_ in their vicinity falls to a critical level of about 1 mm Hg. Thus, in order to meet the energetic needs of cells, it is important to maintain a continuous supply of oxygen to the mitochondria at or above the critical P_O2_. This information is particularly important for cells of the brain and the heart.

Heart myocytes exert mitochondrial OXPHOS life-long and incredibly robust for maximal ATP production. Recent findings demonstrate the pivotal roles of local cardiac myocyte metabolism and ATP-sensitive K^+^ (K_ATP_) channels in regulating blood flow in the heart [24]. Cardiac ventricular myocytes sense the local myocyte metabolic state via K_ATP_ channels and send a negative feedback signal-correction upstream electrically [24]. Another recent study documents crosstalk between mitochondria and the cell’s cytoskeleton in cardiac cells [25]. Mitochondrial positions in cardiac and skeletal muscle cells are tightly fixed, thus, providing their regular arrangement and numerous interactions with other cellular structures. The cytoskeleton with its microtubules, intermediate filaments, and microfilaments plays an important role in the maintenance of mitochondrial shape, location, and motility. Tubulin beta-II participates in the regulation of outer mitochondrial membrane (OMM) permeability through interaction with the VDAC [25].

Mitochondrial Ca^2+^ also plays a critical role in regulating ATP production in the heart. This is due to Ca^2+^-dependent regulation of mitochondrial enzymes of the TCA cycle, of the proteins of the ETC, and of the ATPase itself [26]. Ca^2+^ is transported via the mitochondrial calcium uniporter (MCU) of the IMM from the cytoplasm into the mitochondrial matrix and this transport is driven by membrane potential generated by the ETC. Ca^2+^ influx through the channel pore is tuned by coupled inhibitory and activating sensors on both sides of the IMM [27]. This allows fine-tuning of aerobic metabolism. Mitochondrial Ca^2+^ accumulation stimulates aerobic metabolism by inducing the activity of three TCA cycle dehydrogenases: pyruvate dehydrogenases, isocitrate dehydrogenase, and α-ketoglutarate dehydrogenase [28]. The endoplasmic reticulum chaperon calnexin was reported to control the cellular energy balance between OXPHOS and glycolysis. Calnexin also controls mitochondrial positioning and respiration [29].

A newly discovered mitochondrial potassium channel was inhibited by sodium in a dose-dependent manner. This caused increase in ETC complex IV activity, decrease in complex I activity and ROS production, cell protection, and ATP synthesis [30].

Skeletal muscle represents the largest metabolic organ of the body. These muscles are tissues specialized to generate force to drive body movements. This is achieved through ATP-driven cyclic interactions between the protein filaments actin and myosin. These filaments are organized in a structure called sarcomere. SUMOylation has, in recent years, emerged as a key regulator in sarcomere organization [31]. It was shown that the level of mitochondrial-specific IMM lipid CL oscillates during the diurnal cycle in myotubes [32].

Disorganized sarcomeres lead to human diseases, such as cardiomyopathies and muscle atrophic conditions prevalent in cachexia. Tumor-induced protein breakdown in skeletal muscle involves the two most important cellular degradation systems, the ubiquitin proteasome and autophagy lysosome [33]. More information about cachexia and mitochondria will be dealt with under Section 9.

When the demand for ATP and oxygen exceeds supply, as occurs, for instance, during intense sportive exercise or ischemia, fermentation of pyruvate to lactate increases. Lactate is not only a metabolic waste product, but also serves as a major circulating carbohydrate fuel. Lactate and pyruvate together serve as a circulating redox buffer that equilibrates the NADH/NAD ratio across cells and tissues [34].

### 4.2. Pathophysiologic Situation

Cardiolipin is essential for normal mitochondrial respiratory function. A mouse model of CL deficiency was recently developed by knocking-down (KD) the CL biosynthetic enzyme tafazzin (TAZ KD mice). These mice were shown to develop an age-dependent cardiomyopathy due to mitochondrial dysfunction. Recent results demonstrate that CL deficiency promotes the development of a hypertrophic lipotoxic cardiomyopathy [32]. The dietary antioxidant resveratrol was shown to attenuate the cardiomyopathy by reducing ROS, cardiac steatosis, and by maintaining mitochondrial function [35].

Barth syndrome, a mitochondrial myopathy, results from mutations in the TAZ gene, which encodes a phospholipid transacylase required for CL remodeling. TAZ deficiency was recently shown to impair oxidative metabolism in cardiac mitochondria. Mitochondrial H_2_O_2_ emission and oxidative damage were greater in TAZ KD than in wild type hearts. Fatty acid and pyruvate oxidation capacities were 40–60% lower, but an upregulation of glutamate oxidation supported respiration rates in TAZ KD hearts. The study supports links between CL abnormalities, respiratory supercomplex instability, and mitochondrial oxidant production in the mammalian heart [36].

Cardiovascular diseases are the world’s leading cause of mortality and morbidity. Oxidative stress has effects, not only on muscle cells, but also on endothelial cells. Endothelial dysfunction is defined as a diminished production or availability of nitric oxide (NO). Inflammation-induced oxidative stress results in increased accumulation of ROS, mainly derived from mitochondria. Oxidation of mitochondrial CL loosens cytochrome-c binding. This leads to its release into the cytoplasm and to activation of the apoptotic cascade. A recent review discusses potential diagnostic biomarkers and therapeutic targets of oxidative stress-induced endothelial dysfunction [37].

Mitochondrial calmodulin kinase II has been reported to cause in a murine model adverse metabolic reprogramming and dilated cardiomyopathy [38].

The risk for CVD is increased by hyperlipidemia, hypertension, diabetes, and related disorders. Environment and lifestyle affect epigenetic mechanisms mediated by histone modifications and by long noncoding RNAs (LncRNAs) [39]. Recent studies identified dysregulated histone modifications and chromatin modifying proteins at cis-regulatory elements mediating the expression of genes associated with CVD in vascular and immune cells, in response to growth factors and inflammatory mediators [40]. The role of mitochondria in CVD has recently been investigated. The 17 mtDNA mutations were reported to be associated with ischemic heart disease in humans [40].

Ischemia, a restriction in blood supply to tissues, causes a shortage of oxygen that is needed for cellular energy supply and metabolism. It can affect not only the heart, but also the brain, the bowel, the limbs, and the skin.

New strategies for intervention in CVD: (1) hypoxia-inducible factor-1α (HIF-1α) induces genes that can improve blood circulation. Recombinant adenovirus-mediated HIF-1α secretion was reported to exert a protective effect against ischemia in a rat model [41]. (2) Medical low dose ozone exposure induces a moderate oxidative eustress activating antioxidant pathways [42]. Oxygen-ozone (O_2_-O_3_) therapy has been recommended for patients with peripheral arterial disease (PAD) [43]. (3) Q10 coenzyme supplementation has potential for functional improvement in patients with heart failure [44]. In experimental mouse model studies, Q10 alleviated atherosclerosis [45]. (4) Magnesium supplementation has been proposed for heart failure and CVD because hypomagnesemia is commonly observed in these conditions [46].

## 5. Metabolite Supply for Liver Function and Energy Supply for Nervous System Function

### 5.1. Metabolic and Detoxifying Functions of the Liver

Unlike all other organs, the majority of blood arriving at the liver is venous and is supplied via the portal vein from the intestine. Blood arriving in the liver percolates among Kupffer cells. These cells of the macrophage lineage represent a physical barrier. They are phagocytic and remove particulate material, such as altered red blood cells and bacteria. Hepatocytes exert a second level of defense, which is biochemical. The metabolic detoxification reactions of hepatocytes are divided into two classes. Phase I reactions (oxidation, hydroxylation, and other reactions catalyzed by cytochrome P450 enzymes) are followed by phase II reactions that conjugate the resulting products with another molecule (e.g., glucuronic acid, sulfate, amino acid, glutathione) to promote their excretion.

Hepatocytes contribute to metabolism of the major nutrients, namely carbohydrates, lipids, and proteins. They play an important role in glucose metabolism. By engaging in gluconeogenesis, hepatocytes convert other sugars to glucose. At times of glucose excess, the liver stores glucose as glycogen. It releases stored glucose into the bloodstream when it is needed. Hepatocytes also convert products of carbohydrate metabolism to lipids that can be stored in adipose tissue. The liver participates in lipid metabolism. Lipid metabolic enzymes are engaged in fatty acid oxidation (FAO) to supply energy for other body functions. Hepatocytes also synthesize lipoproteins, cholesterol, and phospholipids. The latter two are important in the biogenesis of cell membranes. Concerning protein metabolism, the liver synthesizes all the nonessential amino acids that do not need to be supplied in the diet. With the exception of immunoglobulins, the liver synthesizes almost all the proteins present in plasma, including albumin and clotting factors for blood coagulation.

Liver regeneration is of particular importance in cases of liver injury or other defects. No other solid organ has such regenerative mechanisms. It is at 100% and based on the liver-to-bodyweight ratio. Experimental models involving partial hepatectomy or chemical injury revealed extracellular and intracellular signaling pathways, which are used to return the liver to size and weight equivalent to those prior to injury. The regenerative activities of hepatocytes and cholangiocytes are typical. When regeneration of one of the two cell types fails, the two cell types function as facultative stem cells and transdifferentiate into each other [47]. Hepatocyte-driven liver regeneration that involves the proliferation of pre-existing hepatocytes is a primary regeneration mode. Another mode is liver progenitor cell (LPC)-driven and involves dedifferentiation of biliary epithelial cells or hepatocytes into LPCs [48].

Liver recruitment of bone-marrow-derived mesenchymal stem cells (MSCs) and their differentiation into adipocytes is a further mechanism of liver regeneration. Hepatic stellate cells (HeSCs) are liver-resident bone marrow-derived MSCs located in the space of Disse. Upon activation, the star shaped HeSCs can differentiate either into myofibroblasts to produce extracellular matrix (ECM), or they can differentiate into adipocytes. It is likely that the Hippo pathway is involved in liver regeneration [49]. Programmed changes in liver glycogen and lipid metabolism were observed during transient graft-versus-host (GvH) and GvL reactivity in a mouse leukemia model system. Five days after immune cell transfer, liver glycogen was depleted. This was promptly followed by lipid recruitment into the HeSCs. Lipid droplets could be visualized in stellate cells by respective staining [50]. Liver regeneration is a complex and well-orchestrated phenomenon, which would not be possible without the help of mitochondria. The process is associated with signaling cascades involving growth factors, cytokines, matrix remodeling, and several feedbacks of stimulation and inhibition of growth related signals [47].

Mitochondrial involvement in liver regeneration has recently been documented: (1) augmenter of liver regeneration (ALR) prevents mitochondrial fission by inhibiting dynamin-related protein 1 (DRP1) phosphorylation, contributing to liver protection. The study showed that the regulation of Drp1 SUMOylation by ALR protects mitochondria from fission and rescues hepatocytes from hepatic ischemic reperfusion injury (IRI)-induced apoptosis [51]. (2) ALR also protects renal tubular epithelial cells from IRI by promoting PINK1/Parkin-mediated mitophagy [52]. (3) The mitochondrial metal transporters mitoferrin1 (Mfm1) and mitoferrin2 (Mfm2) are required for liver regeneration. Loss of both mitoferrins in hepatocytes dramatically reduced regeneration in adult mouse liver [53].

### 5.2. Metabolic Syndrome and Neurodegenerative Diseases

Metabolic syndrome (MetS) is defined as co-occurrence of obesity, insulin resistance, atherogenic dyslipidemia, and hypertension. MetS has been estimated to affect 20–30% of adults of the world [54]. It is associated with increased risk of type 2 diabetes mellitus (T2DM), non-alcoholic fatty liver disease (NAFLD), myocardial infarction, and stroke. The pathogenesis of MetS can be explored from a mitochondrial perspective [54]. Mitochondrial dysfunction contributes to MetS-typical oxidative stress and systemic inflammation.

Mitochondrial oxidative stress can induce hepatic insulin resistance. Treatment of normal human liver cells with insulin for 48 h reduced glucose uptake and increased serine phosphorylation of insulin receptor substrate-1, indicating a reduction in insulin receptor signaling. Such insulin treatment caused (i) accumulation of ROS in the cytosol and mitochondria; (ii) enhanced expression of the antioxidant transcription factor nuclear factor E2-related factor 2 (Nrf2); and (iii) translocation of upregulated Nrf2 to the nucleus. These effects could be ameliorated by cell treatment with the bioactive lipid mediator sphingosine-1-phosphate (S1P) [55].

Insulin resistance (IR) has both heritable and environmental determinants centered on reduced mitochondrial function. Recent studies support the hypothesis that reduced mitochondrial function in insulin-responsive tissues (skeletal muscle, white adipose tissue, liver) is linked to tissue and whole body IR via the effects on cellular energy balance [56]. Mitochondrial metabolite transporters generate unique cytosolic signaling molecules that can regulate insulin release from pancreatic β-cells. Mitochondrial metabolites, transported by the citrate carrier (CIC), dicarboxylate carrier (DIC), oxoglutarate carrier (OGC), and mitochondrial pyruvate carrier (MPC) play important roles in the regulation of glucose-stimulated insulin secretion (GSIS). A recent review discusses a possible link between defective anaplerotic-derived α-ketoglutarate (αKG), hypoxia-inducible factor-prolyl hydroxylases (PDHs), and the development of T2DM [57].

Diabetes mellitus (DM) is caused either by deficiency of insulin (T1DM) or by downregulation of receptors for insulin (T2DM). Insulin is an anabolic hormone from the endocrine pancreas that is secreted in times of excess nutrient availability. It allows the body to use carbohydrates as an energy source and to store these nutrients. Glucagon is a counter-regulatory catabolic hormone to insulin. It is produced and stored in granula in A-type cells of the pancreas. Its secretion increases during periods of food deprivation. It acts to mobilize nutrient reserves. To this end, it mobilizes glycogen, fat, and even protein.

Metabolic diseases are characterized by high NADH/NAD^+^ ratios due to excessive electron supply. This causes defective mitochondrial function and impaired sirtuin-3 (SIRT-3) (an NAD^+^-dependent deacylase) activity with the consequence of driving oxidative stress and altered fatty-acid ß-oxidation [58]. Maintenance of proper NADH/NAD^+^ and glutathione (GSH/GSSG) ratios are important to ameliorate insulin resistance. Alterations in these redox couples lead to electron transport chain complex I dysfunction, disruption of SIRT-3 activity, ROS production, and impaired ß-oxidation. The latter two are key effectors of IR [58].

Good health depends on maintenance of metabolic flexibility. This can be exemplified by pyruvate dehydrogenase enzyme complex (PDC), which plays a central role in carbohydrate metabolism. Regulation of PDC is dependent on rapid changes of its phosphorylation state. This is determined by the relative activities of pyruvate dehydrogenase kinases (PDKs) and pyruvate dehydrogenase phosphatases (PDPs). It was recently reported that overexpression of PDKs in muscle, liver, and the immune system leads to loss of metabolic flexibility [59].

The elimination of dysfunctional and damaged mitochondria by selective autophagy, called mitophagy, is a major mechanism of mitochondrial quality control. Damaged mitochondria can release mitochondrial components, such as mtDNA with hypomethylated CpG motifs or cardiolipin. These molecules are then sensed as danger-associated molecular patterns (DAMPs) and can trigger innate immune signaling. Via release of such DAMPs, mitochondria can guide the immune response towards inflammation. If mitochondrial DAMPs are released by damaged cells, without the presence of an infection, such as after trauma, they can induce undesired inflammatory responses resulting in tissue damage and organ dysfunction [60].

Oxidative stress causes mitochondrial permeability transition (MPT) opening and ATP depletion. Low oxidative damage can be coped with by autophagy and mitophagy thus maintaining cell survival. As oxidative damage increases, cytochrome c is released from mitochondria activating the caspase cascade and triggering apoptosis. Extreme oxidative stress causes rupture of the OMM, irreversible swelling, and necrosis. Infection, injury, and necrosis leads to release of high mobility group box 1 (HMGB1) protein. This DAMP can interact and activate cell surface receptors, such as receptor for advanced glycation end products (RAGE) or toll-like receptor 4 (TLR-4). This leads to nuclear factor kB (NF-κB), mitogen-activated protein kinase (MAPK), and phosphoinositide-3-kinase (PI3K)/protein kinase B (Akt) pathway activation, mediating different responses in hematopoietic and immune cells [61].

Antioxidants such as ethyl pyruvate, quercetin, green tea, *N*-acetylcysteine, and curcumin can be protective partly through attenuating HMGB1 release [61]. Long-term fasting was demonstrated to enhance the endogenous production of antioxidant molecules [62]. Micronutrients (e.g., vitamins and minerals) are also part of the antioxidant system. The importance of nutrients on mitochondrial function has been excellently reviewed recently, in particular with respect to perinatal health [63]. Because of epigenetic imprinting, the right nutrition is very important in the perinatal period. Neonatal brain injury (cerebral ischemia and brain hemorrhage) has been linked to ferroptosis. This is an iron-dependent form of cell death that is characterized by enhanced lipid peroxidation [64].

Symbiosis between bacteria and human beings exists not only in the form of mitochondria. Human microbiota bacteria colonize the gastro-intestinal (GI) tract. They are collected in communities that vary in density and composition with each segment (oral cavity, esophagus, stomach, small intestine, and colon). The gut microbiota is a complex ecosystem in which microbes coexist and interact with the human host. Resident gut bacteria are needed for multiple vital functions, such as nutrient and drug metabolism, the production of energy, defense against pathogens, the modulation of the immune system and support of the integrity of the mucosal barrier [65]. Proton pump inhibitors (PPIs) are common medications in the treatment of acid-related disorders. They act through irreversible inhibition of the hydrogen/potassium ATPase pump. Long-term reduction of gastric acid secretion can increase the risk of an imbalance in gut microbiota composition, i.e., dysbiosis, and of pathogenesis of several specific GI tract diseases [65]. Moreover, antibiotics should be prescribed with care in order not to cause unnecessary damage to the gut microbiota and to mitochondria.

New Strategies for Intervention in MetS: (1) Antioxidants. Antioxidants targeting mitochondrial superoxide anion radicals (^+^O_2_^−^/hydrogen peroxide (H_2_O_2_)) have been proposed for the prevention and treatment of MetS [66]. (2) Molecular hydrogen. Molecular hydrogen (H_2_) can selectively reduce *OH radicals. H_2_ has pluripotent therapeutic potential, including anti-oxidation, anti-inflammation, anti-apoptosis, and anti-shock. The Kelch-like ECH-associated protein-1 (Keap1)-NF-E2-related factor 2 (Nrf2)-ARE signaling pathway (Keap1-Nrf2-ARE), which is activated by H_2_ regulates cellular redox balance, metabolism, and responses against cellular stress [67]. (3) Nanoparticles. Liver-targeted nanoparticles (Nano-MitoPBN) can be sequestered in the liver and cause normalization of glucose metabolism by improving mitochondrial redox balance [68]. They decrease the NADH/NAD^+^ ratio, improve mitochondrial oxidative energy coupling and ATP synthesis, thus alleviating ROS-induced mitochondrial dysfunction [69]. (4) Mitoquinone. Mitochondria-targeted antioxidant mitoquinone was found to attenuate liver inflammation and fibrosis in cirrhotic rats [69]. (5) Targeted peptide. A VDAC1-based peptide (R-Tf-D-LP4) was successfully tested as a potential treatment for diabetes [70]. (6) L-carnitine plus nicotinamide riboside. A combined treatment with L-carnitine and nicotinamide riboside was found to improve hepatic metabolism and to attenuate obesity and liver steatosis [71]. (7) Complex I inhibition. Mitochondrial OXPHOS of liver, but not of muscle cells, was found to increase in obesity and diabetes. It was suggested that only complex I inhibition may ameliorate hyperglycemia via lowering hepatic glucose production [72]. (8) Metformin. Metformin is a first-line anti-diabetic drug. Recent results indicate that metformin not only has glucose-lowering effects. It preserves mitochondrial complex I and membrane potential, reduces ROS production, and increases superoxide dismutase 2 (SOD2). Metformin ameliorated palmitate-induced necrosis and apoptosis in HepG2 cells [73]. (9) Targeting adipocytes. Diabetes medication canagliflozin promoted mitochondrial remodeling of adipocytes via the AMPK-Sirt1-Pgc-1a signaling pathway [74]. Similarly, recombinant DEL-1 attenuated lipid-induced inflammation and insulin resistance in adipocytes via the adenosine-monophosphate-activated protein kinase (AMPK)/homooxygenase 1 (HO-1) pathway [75].

Neurodegenerative diseases: such diseases include the central nervous system (CNS; brain and spinal cord) and the peripheral nervous system (PNS; nerves and ganglia (i.e., small groups of neurons)).

CNS. Certain types of stress-induced mitophagy are regulated by the mitochondrial kinase PINK1 and the E3 ubiquitin ligase Parkin, both of which are linked to autosomal recessive Parkinson’s disease (PD) [76]. Mutations in genes, such as PINK1 and Parkin are responsible for rare genetic forms of PD. Mitochondrial dysfunctions also play a role in other neurodegenerative diseases including Alzheimer’s disease (AD) [77,78]. Selenepezil was found to activate one of the most potent defensive system against oxidative stress, the Keap1-Nrf2-ARE signaling pathway. It alleviated mitochondrial dysfunction and exerted neuroprotective effects in an AD rat model [79].

A particular liver disease is associated with neuropathology. The main cause of hepatic encephalopathy is the accumulation of ammonia (NH_3_) in the brain due to impaired liver detoxification function. Ammonia-induced toxicity in starved rats led to an energy crisis due to decreased ATP synthesis and cessation of gluconeogenesis and ketogenesis. Reduction in glucose and ketone body supply to the brain was a terminal event in liver toxicity, preceding the development of coma [80]. Such studies could provide a framework to further explore the relationship between hepatic dysfunction and progression of a brain energy crisis. Chronic diseases are systemic diseases with multiorgan morbidity. Such diseases are based on dysfunctioning mitochondria. Moreover, 60% of patients show disturbances in the pyruvate dehydrogenase complex. In addition, biochemical diagnostics reveal a pathologic lactate to pyruvate ratio. A rise in NO indicates nitrosative stress, which inhibits ATP synthesis and hem proteins, such as cytochromes and peroxidases [16].

Hippocampal degeneration is associated with a reduction of astrocytes expressing glutamine synthase (GS). In the human epileptic hippocampus, astrocytic GS is downregulated proportionally to the degree of neuronal loss [81]. Perinatal hydrocephalus is associated with a loss of mitochondrial apoptosis-inducing factor (AIF) [82]. Evidence has been presented that Dickkopf-3 (DKK3), a Wnt pathway inhibitor, protects neurons and astrocytes against a variety of toxic insults by inducing vascular endothelial growth factor (VEGF) [83].

Perinatal-life iron deficiency (ID) impairs energy metabolism in a brain region- and age-dependent manner, in particular in hippocampal neurons [84]. Iron-containing proteins involved in brain energy metabolism are aconitase (TCA), succinate dehydrogenase iron-sulfur, and cytochrome b subunits (complex II) (TCA, ETC), NADH dehydrogenase iron-sulfur proteins 1–8 (complex I) (ETC), ubiquinol:cytochrome c oxidoreductase (complex III) (ETC), cytochrome c (ETC), and cytochrome c oxidase (complex IV) (ETC) [84].

Advanced imaging and signal-processing methods allowed identifying redox and pH signaling in neurons at the single mitochondrion level. Such signaling was found to occur particularly in axonal and synaptic organelle clusters [85]. The redox state of the neural progenitors regulates physiological processes, such as neuronal differentiation and dendritic and axonal growth. Thioredoxin (TRX)-related transmembrane-2 (TMX2) is a crucial regulator of cellular redox state. Its dysfunction causes severe brain development abnormalities [86].

PNS. Ischemia-reperfusion injury (IRI) is the tissue damage caused when blood supply returns to tissue after a period of lack of oxygen. It is often associated with microvascular injury, stress and a subsequent inflammatory response. The stress response to peripheral nerve injury was found to extend beyond the injured neuron. The transcription factor forkhead class box O3a (FOXO3a) is a sensor of unilateral nerve injury in sensory neurons ipsilateral, contralateral, and remote to injury. One hour post-injury, an acute increase in nuclear FOXO3a in small-size injured neurons occurred followed by a significant decrease after several days. The response was mirrored in C4 dorsal root ganglion neurons remote to injury [87].

Novel antioxidants, such as mitoquinone and edaravone, have recently been proposed to intervene with oxidative stress induced by traumatic brain injury. MitoQ can cross the blood brain barrier and cellular membranes in the mitochondria and is considered to activate the Nrf2/ARE pathway leading to an increase in the expression of antioxidant enzymes. Edaravone is a free radical scavenger [88].

Mitochondria are highly dynamic organelles. They can be joined by mitochondrial fusion and divided by mitochondrial fission. Cerebral ischemia-reperfusion (I/R) injury can disrupt the balance between mitochondrial fusion and fission causing neuronal death [89]. Another example of mitochondrial dynamics relates to their axonal transport. Mitochondrial Rho GTPases (Miro) are OMM proteins. They play a vital role in microtubule-based mitochondrial axonal transport, mitochondrial fusion and fission, and Mito-Ca^2+^ homeostasis [90]. Miro proteins have been suggested as key players in various neurodegenerative diseases including AD, PD, Amyotrophic lateral sclerosis (ALS) and Huntington’s disease (HD) [90].

## 6. At Work for the Immune System

As cells are stimulated to grow, proliferate, and differentiate or are regulated to rest, their metabolic requirements change. This chapter uses, as example, T lymphocytes. T cell metabolism is at present considered of major importance for T cell function and fate. Regulation of nutrient uptake (e.g., glucose via glucose transporter 1 (Glut1)) and nutrient utilization appears of fundamental importance [4].

### 6.1. Metabolism in Resting Naive and Memory T Cells

Naive T cells continually migrate through secondary lymphoid tissues for immune surveillance. Cytoskeletal rearrangements occur constantly and require ATP. Apart from that, only basal replacement biosynthesis is required. They thus have a metabolic balance that favors energy production over biosynthesis. Resting T cells use predominantly high-energy-yielding processes of pyruvate and glutamine oxidation via the TCA cycle and FAO [91].

When T cells are removed from their in vivo microenvironment, they internalize and degrade Glut1 as well as other nutrient transporters [92]. For maintenance of their viability, two receptor systems are important: The receptor for the cytokine interleukin 7 (IL-7) (IL-7R) [93] and the T cell receptor for recognition of specific antigen (TCR). Signaling through IL-7R occurs via IL-7Rα and the common γ chain. It leads to activation of janus-kinase 3 (JAK3) and phosphorylation of signal transducers and activators of transcription 5 (STAT5), followed by activation of the PI3K/Akt/mammalian target of rapamycin (mTOR) pathway [94].

IL-7 is a pro-survival factor for early thymocytes, resting peripheral naive T cells, and memory T cells (MTCs) [93,95,96]. IL-7 is expressed by stromal cells in T cell zones of lymphoid organs. It is important for preventing atrophy and maintaining resting T cell metabolism. Defects in its pathway are a cause of severe combined immunodeficiency (SCID) in humans and mice [94]. IL-7R regulates glucose uptake mainly through the PI3K/Akt/mTOR pathway [95]. This pathway can promote cell surface trafficking of Glut1. In addition to metabolizing glucose, resting T cells oxidize amino acids [93]. Amino acid transporters are specific targets of IL-7 signaling [95,96,97,98].

Signaling through the TCR also contributes to homeostasis in the naive T cell population. It maintains baseline Glut1 expression. In the absence of TCR signals, Glut1 expression decreases, which leads to metabolic stress and apoptosis [91].

The kinetics of a T cell response consists of three phases: growth and expansion, execution of effector function, and retraction. Usually, a small portion of cells, the MTCs survive. Both, CD4^+^ and CD8^+^ T cells are capable of becoming long-lived MTCs that provide protection against secondary infection or secondary growth of cancer [4,99].

MTCs express high levels of the mitochondrial lipid transporter carnitine palmitoyltransferase 1a (CPT1a). Inhibition of this protein diminished mitochondrial function and reduced memory cell survival [99,100]. Many of the metabolic changes in MTCs are regulated through the induction of mitochondria biogenesis. MTCs have an increased capacity to undergo oxidative metabolism under metabolic stress, a characteristic called “spare respiratory capacity”. Reprogramming of CD8^+^ T cells from a glycolytic back to an oxidative state appears as a key element to allow selection and survival of MTCs [86]. CD8^+^ MTCs use extracellular glucose to support mitochondrial FAO and OXPHOS. It was demonstrated that MTCs rely on cell intrinsic expression of the lysosomal acid lipase to mobilize fatty acids for FAO and MTC development [100].

Tumor necrosis factor receptor-associated factors (TRAFs) are involved in modulation of cell’s life and death [101]. The signaling protein TRAF6 plays a key role for CD8^+^ T cells to switch from glycolytic to oxidative metabolism [102]. TRAF6 is a regulator of NF-κB [102]. TRAF6−/− CD8^+^ T cells had impaired memory responses due to an inability to activate adenosine 3′,5′-monophosphate (AMP)-activated protein kinase (AMPK) and to induce lipid oxidation [103]. AMPK is a metabolic regulator that senses energy deficiency. Its activation promotes ATP-producing metabolic pathways while downregulating ATP-consuming pathways. As a posttranscriptional regulator, AMPK inhibits mTOR and promotes regulatory T (Treg) cells [102].

### 6.2. Metabolic Adaptation upon T Cell Activation

Otto Warburg not only studied the metabolism of cancer cells, but also that of peripheral blood leukocytes. He described that upon mitogenic stimulation of leukocytes, glycolysis and lactate production (i.e., aerobic glycolysis) were strongly increased [103]. Aerobic glycolysis drives pyruvate to lactate, while OXPHOS drives pyruvate into the TCA cycle. The observation was later confirmed with T cells. Antigen-specific stimulation in vivo, a more physiological system than mitogen activation in vitro led to a rapid increase in Glut1 expression [104].

T cell activation involves metabolic reprogramming [105]. This is accompanied by upregulation of Glut1 transporter, increase in glycolysis and lactate production, in OXPHOS and in metabolite supply for anabolic processes. These changes enable cell growth, proliferation, and execution of effector function. Activated CD8^+^ T cells differentiate to provide an antigen-specific cytolytic defense against intracellular pathogens and tumors. CD8^+^ effector T cells are less dependent on Glut1 and oxygen levels compared to their CD4^+^ counterparts [106].

Costimulatory molecules such as CD28 and CD137 (4-1BB) have been reported to be important regulators of metabolic routes in T cells [4,106]. They can modulate glycolysis, mitochondrial respiratory capacities, and FAO [106]. CD28 activation enhances TCR signaling, secretion of IL-2 and expression of its receptor CD25 [106]. This promotes cell proliferation and differentiation into effector cells and MTCs. CD2–CD58 interactions at the outer edge of the immunological synapse, the CD2 corolla, captures CD28 and other costimulatory molecules, and boosts signaling by 77% [107].

Different costimulatory signals and cytokines induce diverse metabolic changes that involve mitochondrial mass and function. Co-inhibitory molecules such as T lymphocyte antigen 4 (CTLA-4) and programmed cell death protein 1 (PD1) on T cells, upon activation, suppress the activity of costimulatory molecules. IL-2, IL-7, IL-9, IL-12, and IL-15 provide further important signals through their respective cytokine receptors for T cell activation. IL-12 and type I interferon, often associated with microbial infections, can provide signal 3 for effective cytotoxic T lymphocyte (CTL) survival and improvement of CD8^+^ effector and memory function [108].

Activated CD4^+^ T cells respond to specific cytokine signals by differentiating into distinct helper T cell (Th) subsets. These contribute to cellular, humoral, and mucosal immunity or suppress T cell activation and excessive inflammatory responses. Table 3 provides an overview of metabolic regulation in T lymphocytes. The protein mTOR is the catalytic subunit of two structurally distinct complexes: mammalian target of rapamycin complex 1 (mTORC1) and mTORC2. mTORC1, which regulates Th1 cell functions, is a nutrient/energy/redox sensor and controls protein synthesis. The activity of mTORC1 is regulated by rapamycin, insulin, growth factors, phosphatidic acid, certain amino acids, mechanical stimuli, and oxidative stress. mTORC2 functions as regulator of the actin cytoskeleton through its stimulation of F-actin stress fibers, paxillin, RhoA, Rac1, Cdc42, and protein kinase Cα (PKCα). mTORC2 also exhibits tyrosine protein kinase activity and phosphorylates the insulin-like growth factor 1 receptor (IGF-IR) and insulin receptor (InsR) leading to their full activation. Disruption of mTORC2 produces the diabetic-like symptoms of decreased glucose tolerance and insensitivity to insulin. mTORC2 is a molecular determinant of Th2 differentiation [4].

Th9 cells represent a relatively new subset of CD4^+^ T cells well suited for adoptive cell therapy of advanced tumors. They produce interleukin-9 (IL-9), are cytolytic and hyperproliferative with long persistence in vivo without exhaustion. Their unique molecular determinants are upregulated expression of Traf6 and Eomes and activation through the pathway Pu.1-Traf6-NF-κB [109]. Th17 cells are characterized by the molecular determinants mir26 [110], HIF-1, pyruvate kinase M2 (PKM2) [111] and adiponectin [112]. Treg cells (Foxp3+) have a suppressive function with respect to autoimmune reactivity [113]. In addition, these cells, via epigenetic regulatory mechanisms, maintain self-tolerance and support organ homeostasis [114]. The transcription factor basic leucine zipper ATF-like TF (BATF) was just identified for precursors of Tregs in non-lymphoid tissue [115].

T cell metabolism is regulated at the transcriptional and at the post-transcriptional level. Examples of transcriptional regulators are HIF-1α, liver X receptor (LXR), estrogen-related nuclear receptor (ERRα), c-Myc, and Notch. In short, HIF-1α upregulates glycolytic genes and promotes Th17 generation. LXR downregulates T cell activation and upregulates cholesterol and lipid efflux. ERRα upregulates mitochondrial oxidation. The oncogenic transcription factor c-Myc upregulates glycolysis, glutamine metabolism, and cell cycle genes. Notch (a signaling receptor of importance in embryogenesis) promotes glucose metabolism during T cell development [4].

Examples of posttranscriptional regulators are extracellular signal-regulated kinases (ERK), AMPK, mTOR and phosphoinositide-dependent kinase 1, which phosphorylates Akt (PDK1/Akt). ERK upregulates glutamine uptake. AMPK inhibits mTOR and promotes Treg cells. mTOR upregulates Glut1 surface expression and promotes effector T cells. PDK1/Akt promotes Glut1 surface expression [4].

In pro-inflammatory cells, such as activated monocytes, T and B cells, the energy is generated by increasing glycolysis. In contrast, in regulatory cells such as Treg or M2 macrophages, energy is generated by increasing mitochondrial function and FAO [4].

A link has been suggested between immune signaling via NF-κB and redox sensitivity of the adapter myeloid differentiation primary response gene 88 (MxD88). Upon oxidation by H_2_O_2_, MyD88 forms distinct disulfide-linked conjugates, which are reduced by the MyD88-interacting oxidoreductase nucleoredoxin (Nrx). Such thiol switches have physiological relevance in mitochondria [116].

### 6.3. Pathophysiologic Situations

Regulation of the balance between oxidative and glycolytic cell metabolism is also important with regards to T cell anergy (e.g., absence of costimulatory signals) and exhaustion (chronic T cell stimulation) [4]. T cell exhaustion can lead to premature senescence. Examples of dysregulations within the immune system are anti-self-reactivity (autoimmunity) and over-reactivity (e.g., cytokine storm).

New insights into immune pathology (1–3) and immune system disorders (4 and 5): (1) T cell anergy. Anergic T cells fail to upregulate the amino acid transporter CD98 and the transferrin receptor CD71 [117]. Ligation of programmed cell death protein 1 (PD-1) can inhibit both mTORC1 and mTORC2 signaling, thus diminishing glycolytic metabolism and contributing to exhaustion [118]. (2) T cell exhaustion and intervention: skeletal muscle activity (i.e., sportive exercise) was reported to be able to antagonize antiviral CD8^+^ T cell exhaustion [119]. Hydrogen gas was reported to activate Q10 to restore exhausted (PD-1+ Tim3+) terminal CD8^+^ T cells. This led to longer overall survival in lung cancer patients treated with nivolumab [120]. (3) Premature senescence. T cells with dysfunctional mitochondria induce multimorbidity and premature senescence. An important role in this phenomenon plays the mitochondrial transcription factor A (TFAM). T cells, thus, are important, among others, to regulate organismal fitness and life span [121]. (4) Autoimmunity. T cell metabolism also plays an important role in diseases like autoimmunity (e.g., systemic lupus erythematosus (SLE), graft-versus-host disease (GvHD), and rheumatoid arthritis (RA)) [4]. Th17 cells (Table 3) were recently demonstrated to play an important role in RA via microRNA 26 [109], pyruvate kinase M2 (PKM2) [112], and the hormone adiponectin [116]. (5) Over-reactivity. The “cytokine storm” syndrome is a pro-inflammatory over-reaction of the immune system, often in response against a viral pathogen. Dysregulated retinoic acid inducible gene-I (RIG-I) responses involving type I interferon via mitochondrial antiviral signaling (MAVS) [122] and viral escape mechanisms [123] are likely involved. This holds true, in particular, for responses to cytosolic RNA viruses, including Ebola [123] and SARS-CoV-2 [124]. Mitochondria may circulate “cell-free” in non-nucleated platelets and extracellular vesicles so that their DAMPs can induce innate immune responses leading to a coagulation cascade [124]. HIF-1α induced changes in monocytes in the lungs of COVID-19 patients. These changes directly inhibited T cell responses and reduced epithelial cell survival [125].

In conclusion, cell metabolism is a key regulator of T cell function and fate. Tight immunometabolic control is of importance. In similarity to cancer cells, activated T cells undergo various metabolic changes that allow them to grow and proliferate rapidly. In this process, T cells prioritize on efficiency in biosynthesis over energy generation.

## 7. At Work in the Bone Marrow for Hematopoiesis and Maintenance of Immunological Memory

### 7.1. Hematopoiesis

Hematopoietic stem cells (HSCs) can be in states of quiescence or self-renewing and yet can differentiate into multiple blood cell types. HSCs reside in a hypoxic niche within the bone marrow (BM). Such perivascular micro-niches are established by CXCL12-abundant-reticular (CAR) cell subsets (Adipo-CAR and Osteo-CAR), which differentially localize to sinusoidal and arteriolar surfaces and act locally as professional cytokine-secreting cells [126].

HSCs rely on glycolysis while differentiated and committed progenitors rely on OXPHOS [127]. S100A6 is a critical HSC regulator. S100A6 knockout (KO) HSCs showed decreased levels of phosphorylated Akt (pAkt) and heat-shock protein 90 (Hsp90). These KO cells revealed impairment of respiratory capacity and a reduction of mitochondrial calcium level [128]. Thus, pAkt was shown to be the prime downstream mechanism of S100A6 in the regulation of HSC self-renewal by specifically controlling mitochondrial metabolic function and Hsp90 protein quality. A putative role of NF-κB signaling but not of HIF-1α has been proposed as a hypoxia-induced regulator in early hematopoietic cells [129].

Tuberous sclerosis complex 1 (TSC1) is a tumor suppressor gene that inhibits cell growth via mTORC1. TSC1 modulates self-renewal and differentiation in HSCs. Recently, it was shown that TSC1 also regulates the proliferation capacity of BM-derived mesenchymal stromal stem cells (MSCs) which are multipotent stromal cells. The data support the view of a pivotal role for the TSC1-mTOR axis in the maintenance of the mesenchymal progenitor pool [130]. There is growing evidence that MSCs exert antioxidant properties [131]. Their modulation of the redox environment and oxidative stress can mediate anti-inflammatory and cytoprotective effects [131].

A crosstalk between HSCs and the BM microenvironment appears critical for homeostasis and hematopoietic regeneration. Intercellular signals by the BM stromal cells regulate HSC localization, maintenance, and differentiation within BM HSC niches. Connexin Gap junction channels (GJ) exert cell-intrinsic and cell-extrinsic effects on HSC and BM stromal cells. They control the transfer of ions, small metabolites, and organelles to adjacent cells [132]. Recent findings reveal a novel function of mitochondria in directly contributing to cellular reprogramming [133]. Treatment of insulin-producing cells from adult peripheral blood (PB-IPC) with platelet-derived mitochondria was shown to reprogram the transformation of PB-IPC into functional CD34^+^ HSC-like cells, leading to the production of blood cells, such as T- and B-lymphocytes, monocytes/macrophages, granulocytes, red blood cells, and megakaryocytes [134].

BM transplantation frequently requires host irradiation pre-conditioning to ablate endogenous hematopoietic cells. In allogeneic GvL situations, host irradiation is also necessary to prevent HvG reactivity. It was recently reported that irradiation damages BM MSCs mitochondrial function. Healthy donor hematopoietic stem and progenitor cell (HSPC) transfer facilitated transfer of functional mitochondria to the stromal BM microenvironment. Such transfer was cell-contact dependent and mediated by HSPC connexin-43 (Cx43) [134]. These results demonstrate that HSPC transfer not only reconstitutes the hematopoietic system following transplantation, but also supports and induces the metabolic recovery of irradiation-damaged microenvironment via mitochondria transfer [131,132,133,134].

The fact that mitochondria function systemically, and permit bioenergetic crosstalk, can be well illustrated by intercellular transfer from MSCs to various ocular cells. Intercellular transfer of mitochondria occurred through F-actin-based tunneling nanotubes [135]. Photoreceptors of the retina, particularly of cones, have high energy demands to mediate color vision. The highest density of ATP generating mitochondria is in photoreceptor cells. Mitochondria, such as chloroplasts, contain pigments with specific light absorbance characteristics. Specific wavelengths are being employed to enhance mitochondrial performance to moderate the aging process [136]. Carotenoid homeostasis is controlled by the mitochondrial enzyme β-carotene-oxygenase 2 (BCO2) and plays an important role in the pathway for vitamin A production. Vitamin A deficiency is associated with age-related macular degeneration. BCO2 was just demonstrated to exhibit catalytic activity towards carotenoids and apocarotenoids [137].

Protein synthesis has been described to play a pivotal role in neural stem cell activation [138]. A tight balance between the quiescence and activation state is instrumental to meet the demands of the brain for a specific cell type at the correct numbers, at a given time and position. The switch between a quiescent and activated state crucially depends on repression or activation of protein synthesis, one of the most energy-demanding processes of the cell. The excellent review describes in particular the role of RNA binding proteins (RBPs) and miRNAs. RBPs are involved in alternative splicing, RNA processing, nuclear export, mRNA stability, and translation. In highly polarized cells like neurons, RBPs help to locally and spatially regulate translation efficiency and successfully integrate environmental signals [138].

### 7.2. Maintenance of Immunological Memory

Memory T cells, upon cognate antigen re-activation, show a faster response compared to naive T cells and the response is polyfunctional in terms of cytokine secretion. MTCs express the key survival receptors IL-7Rα and IL-15Rβ. For long-term survival, MTCs, expressing the chemokine receptor CXCR4, are recruited via the corresponding chemokine CXCL12 (SDF-1α/β) to special niches in the BM. These provide, among others, the corresponding pro-survival cytokines IL-7 and IL-15. The BM maintains memory CD4^+^ T cells to systemic antigens by non-follicular re-activation in immune cell clusters. Following expansion, the cells are recruited into the periphery to the antigenic site for execution of their specific function. The response lasts for about 10 days. Thereafter, the MTCs return to their niches where they are maintained as resting cells [139].

Dynamics and longevity of an MTC response has been studied in a model system in which the bacterial lacZ gene product ß-galactosidase (Gal) served as a surrogate tumor antigen (TA). The first CD8^+^ T cell response was observed after inoculation of live lacZ transfected ESb (ESblacZ) tumor cells into syngeneic DBA/2 mice at a site, the ear pinna, where they could not grow. The first Gal-specific CD8^+^ T cell response was observed in the BM with a peak after 10 days. T cell priming in the BM via antigen-presenting dendritic cells (DCs) had been demonstrated in this model system for the first time worldwide [140]. Following T cell priming in the BM, intra-peritoneal (i.p.) challenge with irradiated ESblacZ cells caused recruitment of Gal-specific MTCs from the BM to the peritoneal cavity. This secondary response involved an expansion of Gal-epitope specific CD8^+^ T cells by a factor >80. Two months later, the MTCs had returned into a resting state and back into the BM. The metabolic reprogramming of such a response can hardly be imagined without the help of mitochondria for provision of energy and metabolites [141].

To investigate the longevity of BM MTCs, the above-described response was transferred to athymic nude (*nu*/*nu*) mice as secondary host, without endogenous T cells. The ins and outs of the transferred MTCs from the BM to the peritoneum and back could be followed up to the 6 antigen restimulation contact. This occured in quaternary nude hosts. Nude mice that were transferred with effector MTCs rejected a tumor dose of 5 × 10^7^ cells and they survived longer than 8 months. In contrast, naive nude mice, upon injection of 1 × 10^5^ respective tumor cells died within 10 days [140].

One subtype of MTC is a stem cell-like MTC (S MTC). This type makes up a substantial proportion of BM MTCs; it is characterized by CD69 and CD127 expression, and by efflux capacity. The S MTCs are strongly resistant to immune tolerance induction, and are, thus, of great interest. An in vitro co-culture procedure has recently been described to convert activated T cells into S MTCs. The stimulating cells were OP9 stroma cells expressing Notch ligand [142]. IL-7 and IL-15 have been reported to be capable to instruct the generation of S MTCs from naive precursors in vitro [143].

Maintenance of immunological memory has also been studied with respect to the consequences of starvation. Collins et al. recently reported that the BM protects and optimizes immunological memory during dietary restriction (DR) [144]. Under DR, MTCs accumulated within the BM but collapsed in secondary lymphoid organs. In response to nutritional stress, glucocorticoids contributed to profound remodeling of the BM. To adopt a state of energy conservation, adipogenesis, and erythropoiesis was activated. Fatty acids were mobilized to be degraded in mitochondria via ß-oxidation. To recruit MTCs from peripheral tissues to the BM, T cell homing factors were upregulated. This involved interactions of CXCR4 with its ligand CXCL12 and of sphingosine-1 phosphate receptor S1P_1_R with its ligand S1P [140]. Of further interest is that in the context of energy supply for the immune system, it was recently discovered that human adipocytes from the BM display distinct immune regulatory properties [145].

In conclusion, the BM exerts many functions of importance for survival of vertebrates, including human. The maintenance of HSCs serves to reconstitute red and white blood cells after damage (e.g., by physical trauma or irradiation) and the maintenance of MTCs serves to protect the host against recurrent infections and cancer. The BM, thus, is unique and combines primary and secondary immune functions [140,141].

## 8. Mitochondria Serving for Cancer: Metabolic Dysregulation

### 8.1. Tumor Cell Metabolic and Genetic Mechanisms

Cancers can be considered as diseases of dysregulated physiology [146]. Dysregulation may occur at different levels: DNA (genetics and epigenetics), RNA (in particular regulation via miRNA), intra-cellular organelles (in particular mitochondria), the plasma membrane and the glycocalyx, the ECM, the cells microenvironment, and systemic systems. During cell proliferation, glucose is an excellent energy source. The oncogene KRAS induces glucose uptake through upregulation of the Glut1 transporter [147].

Metabolic reprogramming in cancer makes the cells minimally dependent on extrinsic stimuli. This is achieved by deregulation of pathways related to energy production and biosynthesis [147]. The main dysregulatory mechanisms of cancer concern (1) aerobic glycolysis of glucose; (2) a “truncated” TCA cycle; (3) activation of cellular metabolism via the PI3K pathway [136]; and (4) lipid synthesis and FAO [147].

(1) Cancer cells employ aerobic glycolysis, the “Warburg effect”. It quickly converts glucose into lactate, generates cytosolic ATP, and leads to robust biosynthesis of proteins, lipids, and nucleotides. (2) In tumor cells, the TCA cycle is used mainly to generate intermediates for biosynthetic purposes. A high flux of intermediates out of the pathway prevents it from acting as a true cycle. Tumor cells, thus, use a “truncated” TCA cycle. Glutamine oxidation makes the truncated TCA cycle possible. (3) During cell division, de novo nucleotide biosynthesis is highly induced. Tumor cells require for this complicated process carbon and nitrogen from multiple sources. The most prominent intracellular regulator of cell growth is mTOR and its network, including receptors for growth factors (e.g., epithelial growth factor (EGF), platelet-derived growth factor ( PDGF), vascular endothelial growth factor (VEGF), insulin/insulin-like growth factor 1 (IGF1). (4) Lipid synthesis requires cytosolic acetyl-CoA, which is mainly derived from mitochondrial citrate of the TCA cycle. FAO comprises a cyclical series of reactions that results in the shortening of fatty acids via β-oxidation.

Elucidation of the metabolic plasticity of cancer revealed mitochondrial reprogramming and hybrid metabolic states. Three stable metabolic phenotypes could be acquired: (1) glycolytic (high HIF-1 and low AMPK); (2) OXPHOS (low HIF-1 and high AMPK); and (3) hybrid (high HIF-1 and high AMPK) [148].

Genetic mechanisms (mutations, amplifications, fusions) behind the metabolic transformation in tumors that activate PI3K or its effectors, affect genes, such as Her2/neu (e.g., breast cancer), breakpoint cluster region-Abelson murine leukemia viral oncogene homolog (BCR-ABL) (e.g., chronic myeloid leukemia (CML), acute lymphoblastic leukemia (ALL), phosphatase and tensin homolog (PTEN) (e.g., glioma) and Akt2 (e.g., ovarian cancer) [147,149]. Other common mutations affect the Myc gene [150]. This is a master regulator of cell cycle entry and proliferative metabolism. The transcription factor c-Myc stimulates proliferation in part by activating expression of several of the cyclins and cyclin-dependent kinase 4 (CDK4), which promote entry into the S phase. c-Myc’s target genes include enzymes involved in nucleotide and 1-carbon metabolism, without which cells could not successfully complete the S phase [147,150].

The role of mtDNA variations in cancer development has recently been systematically investigated. Polymorphisms were found to occupy first rank among mtDNA variations [151]. Mitochondrial metabolites such as ATP, acetyl-CoA. NADH, and ROS, have the potential to alter the metabolism-epigenome-genome axis [152]. Mitochondrial 1-carbon metabolites such as methyl and acetyl groups are primary precursors for epigenetic modifications in eukaryotic cells. Substrate availability regulates the crosstalk between mitochondrial function and the epigenome, and is mediated by energy as well as redox metabolites. This is critical for chromatin remodeling and gene expression, during differentiation and embryogenesis but also in tumorigenesis. As an example, treatment of neuroblastoma cells with a 5-aza-2′-deoxicytidine DNA methyltransferase inhibitor (decitabine) could provide a therapeutic advantage. It downregulated mitochondrial genes related to redox homeostasis [153].

The significance of mitochondrial dysfunction in cancer has recently been reviewed. It includes TCA cycle enzyme defects, mtDNA mutations, defective electron transport chain, oxidative stress or aberrant oncogene and tumor suppressor gene (e.g., TP53, PTEN) signaling [154]. Some early phase I clinical trials in pancreatic cancer (PDAC) target OXPHOS or the TCA cycle by mitochondrial metabolism inhibitors such as metformin or devimistat [155].

### 8.2. Cancer Stem Cell Metabolism

Cancer stem cells (CSCs) are considered a main cause for tumor recurrence. Chronic myeloid leukemia (CML) results from transformation of HSCs by the BCR-ABL kinase. Recently, it was described that silent information regulator 1 (SIRT1) regulates metabolism and leukemic potential in CML stem cells. This NAD^+^-dependent deacetylase deacetylates many proteins, including histones and transcription factors. SIRT1 was found to mediate increased OXPHOS in CML leukemic stem cells [156]. Increased OXPHOS was also required for maintenance of stemness in liver cancer stem cells (LCSCs) [157].

Acute myeloid leukemia (AML) is hierarchical with leukemic stem cells and progenitor cells giving rise to mature and differentiated blasts. Like HSCs, AML cells have unique mitochondrial and metabolic features with reliance on OXPHOS [158]. Recently, the mitochondrial transacylase tafazzin (TAZ) was demonstrated to regulate AML stemness by modulating intracellular levels of phospholipids [159]. This X-linked gene was identified by a clustered regulatory interspaced palindromic repeat (CRISPR) screen. TAZ is the enzyme responsible for cardiolipin remodeling. The CRISP technology receives the Nobel Prize for chemistry 2020. Interestingly, the CRISPR/Cas9 technology was invented by bacteria in the fight against virus (bacteriophage) infection.

New strategies of intervention with cancer stem cell mitochondria metabolism: (1) Downregulation of OXPHOS. Mitochondrial division inhibitor 1 (Mdivi-1) reduced the level of OXPHOS, prevented the collapse of mitochondrial membrane potential and ATP depletion and suppressed the occurrence of autophagy. In contrast, 2-deoxy-D-glucose enhanced cancer stemness by upregulating OXPHOS [160]. (2) Targeting ROS and reactive nitrogen species (RNS). Mitochondria play an important role in CSC resistance to cytotoxic drugs or radiotherapy [161]. The whole mitochondrial cycle, from its biogenesis to mitophagy or apoptosis can be targeted to reduce mitochondrial fitness in cancer. Two elements are usually boosted, namely an augmentation of RNS and ROS and activation of the intrinsic apoptotic pathway [160,161].

### 8.3. The Tumor Microenvironment

Solid tumors are characterized by low glucose concentration, hypoxia, and low pH. These metabolic conditions are a problem for tumor-infiltrating T cells (TILs) by limiting nutrient availability and energy efficiency. Artificial modulation of glycolysis, mitochondrial respiratory capabilities, and fatty acid β-oxidation are known to enhance antitumor performance. Immune escape is another critical hallmark in solid tumors [162]. Tumor cells hijacking checkpoint receptors (e.g., CTLA-4, PD-1) cause deficiency of innate immune sensing and CTL activity. Immune checkpoint inhibitor (ICI) therapy has opened a new strategy to treat cancer. Much further research is required for better understanding the immune ecosystem under ICI therapy, for proper adjuvants and dose calculation to increase desired responses and reducing severe adverse events [161,162]. Cancer vaccines and oncolytic viruses (OVs) exert much lower side effects than other systemic therapies, including ICI [163].

### 8.4. Strategies for Intervention with the TME

Use of nanoparticles (NPs). NPs are presently developed as a new means of interfering with metabolic dysregulation in cancer. A lipid nanoparticle (LNP) was used to target and deliver siRNA to M2-type tumor-associated macrophages (TAMs) [164]. A therapeutic response was obtained through the silencing of the signal transducer and activator of transcription 3 (STAT3) and of HIF-1α. This caused an increase of CD11b^+^ cells and CD169^+^ M1 macrophages in the TME [165]. A second example are dual stimuli-responsive NPs. These have a multilayered structure: a protein cage with small heat shock proteins, a cationic layer, and a dual-sensitive coat. The release of the drug paclitaxel was possible only under conditions similar to the TME, e.g., acid pH and overexpressed hyaluronic acid [165]. A third example are nanoliposomes loaded with isoliquiritigenin to affect AMPK/mTOR mediated glycolysis in colorectal cancer [166].

Activate tumor suppressor genes. The tumor suppressor PTEN drives an “anti-Warburg state”. Less glucose is taken up and it is more efficiently directed towards the TCA cycle. This has implications for the immune system and the TME [167].

Reverse immune suppression. (i) Extracellular adenosine (ADO), present at high concentrations in the TME, suppresses natural killer (NK) and T cell function. Intratumoral generation of ADO depends on the sequential catabolism of ATP by two ectonucleotidases: CD39 and CD73. Inhibition of CD73 by the small molecule inhibitor AB680 was recently demonstrated to be able to reverse ADO-mediated immune suppression [168]. (ii) Adenosine A_2A_ receptor antagonists are being developed as new checkpoint inhibitors for cancer immunotherapy [169]. (iii) Another potentially new checkpoint target could be neuropilin-1 (NRP1). Recent data suggest that NRP1 acts as a barrier to the long-term durability of CD8^+^ T cell-mediated immunosurveillance [170]. (iv) Tumor-derived small extracellular vesicles (sEVs) can induce immunosuppressive Treg cells. In case of sEVs derived from pancreatic ductal adenocarcinoma (PDAC), T cell treatment with sEVs induced overexpression of immune checkpoint proteins PD-1, PD-L1, CTLA4, and Tim-3 and the enrichment of FOXP3+ Treg clusters. ATM, AMPK, SIRT1, SIRT2, and SIRT6 were activated sequentially in sEVs-treated T cells and essential for sEVs-upregulated expressions of FOXO1A, FOXO3A, and FoxP3. These new findings may provide new immunotherapy strategies for PDAC treatment [171].

Uncoupling FOXO3A mitochondrial and nuclear functions. FOXO3A is a transcription factor that plays a key role in cancer cell homeostasis. It plays a dual role in survival/death response to metabolic stress and cancer cytostatic therapeutics. FOXO3A is recruited to mitochondria through activation of MEK/ERK and AMPK. Once imported, it is cleaved to reach mitochondrial DNA. Normal cells under metabolic stress require only the AMPK signal on S30 to direct FOXO3A into the mitochondria. ERK involvement in FoxO3A mitochondrial localization seems exclusive to cancer cells. In cancer cells treated with chemotherapeutic drugs, MEK/ERK dependent accumulation of FoxO3A into mitochondria promoted cell survival. Mitochondrial FOXO3A was required for apoptosis induction by metformin [172]. In patients with EpCAM-expressing tumors, treatment with the EpCAM-neutralizing antibody EpAb2-6 might uncouple FOXO3A mitochondrial and nuclear functions. In murine studies, in vitro treatment with this antibody inhibited AKT and FOXO3A phosphorylation, increased its nuclear translocation, and promoted tumor cell apoptosis. In vivo, its application in an orthotopic model of human colorectal cancer extended overall survival [173].

Target tumor-associated macrophages (TAMs). Brachytherapy in a single dose of 10 Gy can affect the TME by reducing the accumulation of pro-tumorigenic M2 polarized TAMs and by increasing CD8^+^ CTLs [174]. Low dose ionizing radiation (LDIR) programs macrophage differentiation to an inducible nitric oxide synthase (iNOS) positive M1 polarized phenotype that can orchestrate effective T cell immunotherapy, as demonstrated in a xenotransplant model of human pancreatic carcinoma [175]. LDIR is capable of utilizing known molecular master switches such as Nrf2/keap1 or NF-κB/IκB to mediate hormetic effects by H_2_O_2_ signaling [176]. Findings in redox biology revealed that H_2_O_2_ and other ROS species can act as pleiotropic physiological signaling agents [177,178].

Target glutamine metabolism. Glutamine metabolism can be targeted by small molecule inhibitors. These can inhibit myeloid-derived suppressor cells (MDSCs) and enhance tumor-specific immunity [179].

Induce immunogenic cell death (ICD) by radiotherapy. Heat shock proteins (HSPs), high mobility group box 1 (HMGB1), ATP, and its metabolites such as adenosine are the most common DAMPs that are released after radiotherapy-induced ICD [180].

Induce ICD by oncolytic viruses (OVs). A new strategy of inducing ICD and cancer immunotherapy combines hyperthermia and oncolytic virus (OV) pretreatment with specific autologous antitumor vaccination [181]. As OV serves Newcastle disease virus (NDV), which is being applied to cancer patients since more than 50 years [182]. It has the potential not only to induce cancer selective ICD and to stimulate innate and adaptive immunity cells but also to break therapy resistance [183]. NDV targets the oncogenic protein Rac1 and promotes virus replication via syncytia, autophagy, and exosomes. It has the potential to break T cell tolerance towards TA expressing tumor cells, to break resistance to chemo- and radiotherapy, to break resistance to apoptosis and to hypoxia and to break resistance to immune checkpoint blockade and to anti-viral immunity [183].

Target tumor-associated dendritic cells (DCs). DCs in the TME play an important role. Evidence was recently provided of a plasmacytoid DC subset in the TME that favors antitumor immunity [184]. These pDCs expressed high levels of OX40 in the TME of head and neck squamous cell carcinoma. OX40^+^ pDCs produced type I interferons, exerted a distinct immunostimulatory phenotype, cytolytic function, and ability to synergize with conventional DCs (cDCs) in generating potent TA-specific CD8^+^ T cell responses [184]. Induction of systemic immune responses and reversion of immunosuppression in the TME was recently reported by a therapeutic vaccine for cervical cancer [185]. The vaccine contained the human papillomavirus (HPV) HPV16 E7 43-77 peptide and the adjuvant unmethylated cytosine-phosphate-guanosine oligodeoxynucleotide (CpG). The vaccine induced a significant increase in splenic IFN-γ-producing CD4^+^ and CD8^+^ T cells as well as TILs. Moreover, marked decreases of suppressive cells (MDSCs, Tregs, M2 macrophages) were observed in the vaccine group. The vaccine reversed a tolerogenic and immunosuppressive TME and induced robust antitumor immune responses, which resulted in the inhibition of established implanted tumors [185]. Similarly, DC vaccination and CD40-agonist combination therapy licensed T cell-dependent antitumor immunity in a pancreatic carcinoma murine model [186]. Intratumoral injection of the seasonal flu vaccine also caused reprogramming of the TME. It turned “cold” tumors, which were not immune cell infiltrated into immune cell infiltrated “hot” tumors, which have a better prognosis [187].

Results from a meta-analysis. Efficacy and safety of DC vaccines have been evaluated for patients with glioblastoma multiforme (GBM) involving 6 phase II randomized-controlled studies with 347 patients. DC vaccine was associated with significantly improved OS (*p* = 0.03) [188].

## 9. Mitochondria and Cancer Cachexia

### 9.1. New Insights

Cancer cachexia is a multifactorial syndrome characterized by a progressive loss of skeletal muscle mass. It is associated with adipose tissue wasting, systemic inflammation and other metabolic abnormalities. Some related conditions, like sarcopenia (age-related muscle wasting), anorexia (appetite loss) and asthenia (reduced muscular strength and fatigue), share some key features [189].

New recent insights into cachexia are summarized in Table 4.

Mitochondrion metabolism. Evidence was presented implicating disturbed muscle mitochondrial OXPHOS proteome and NAD^+^ homeostasis in experimental cancer cachexia [188]. Another recent study revealed altered mitochondrial metabolism and suppressed acyl-CoA synthase-1 (ACSL1) in mice with colon-26-induced cachexia [191]. Altered hepatic mitochondrial function with suppressed ACSL1 appeared associated with cachexia severity.

Receptor signaling pathways. The SIRT1-NOX4 signaling axis was proposed to regulate cancer cachexia [190]. Rescuing SIRT1 expression reverted myotube wasting. Knocking out Nox4 in skeletal muscles abrogated tumor-induced cachexia in mice [192]. Targeting RAGE prevented muscle wasting and prolonged survival in cancer cachexia [193].

Transcription factors. A regulatory role was suggested for the transcription factor Twist family basic helix-loop-helix transcription factor 1 (Twist1) [194].

Regulatory RNAs. microRNAs (miRNAs) and non-coding RNAs (ncRNAs) have been reported to be involved in cachexia at the sites of skeletal muscle [195,196,197,198].

Hormones. Hormones also play an important role in cachexia. Hindbrain growth differentiation factor 15 (GDF15) is a hormone conveying somatic distress to the brain [199]. GDNF family receptor alpha like-RET proto-oncogene (GDF15-GFRAL-RET) receptor antagonism is emerging as a therapeutic strategy for anorexia/cachexia syndromes [200]. Fat storage depletion from adipose tissue is facilitated by several adipokines [201], which are peptide hormones, such as asprosin [202], leptin [203], and intelectin-1 [204].

Cytokines. Infiltration of adipose tissue by CD14^+^ monocytes leads to increased production of interleukins (e.g., IL-6) and chemokines (e.g., MCP1) and plays an important role in the regulation of metabolism of glucose and lipids [205]. A systematic review and meta-analysis of 20 studies with inhibitors of the IL-6 signaling pathway suggests that this pathway is involved in regulation of body weight [206].

Immune cells. Immune cells also play a role in cancer cachexia. Circulating neutrophils invaded the central nervous system via chemokine (CCL2)-chemokine receptor 2 (CCR2) mediated cachexia during pancreatic cancer [207]. An elevated neutrophil-to-lymphocyte ratio was found to be associated with weight loss and cachexia in advanced colon, lung, or prostate cancer [208]. A negative correlation was observed between CD163^+^ macrophage infiltration and STAT3 signaling in skeletal muscles and pancreatic cancer cachexia [209].

Glucans. The glucan botryosphaeria significantly reduced tumor growth, body-weight loss and cachexia in obesity rats inoculated with Walker-256 tumor cells. Such treatment decreased mesenteric fat and insulin resistance, corrected macrocytic anemia, and increased FOXO3A activity [210].

### 9.2. Intervention with Cancer Cachexia: Immune Counter Attack!

A mouse tumor model has demonstrated that it is possible to reverse cancer cachexia [211]. It is a model of GvL by transfer of allogeneic MHC-matched immune activated T cells. Cachectic tumor-bearing mice had local tumors of 1.5 cm diameter and macroscopic liver and kidney metastases. Reversion of cachexia and complete remission of cancer in late-stage disease had been possible by a combination of sublethal (5 Gy) irradiation followed by a single intravenous transfer of 20 million antitumor immune spleen cells from tumor-immunized resistant mice.

Two to three days after treatment, reversion of tumor tissue pH from acid to neutral was a first sign of this immunotherapy. High field nuclear magnetic resonance (^31^P-NMR) of phosphor-monoesters (PME) and Pi of spectra acquired from tumors before and after adoptive transfer of immune T cells revealed this effect [212].

Five days after treatment, changes in liver glycogen and lipid metabolism were observed [50].

Six days after transfer, CD4^+^ T cells could be seen in the tumors. Moreover, a broad capsule of fibrous tissue between the tumor and the skin could be seen in which Langerhans cells and dermal DCs were embedded [213].

Six to twelve days after transfer, a massive infiltration of liver metastases by donor CD4^+^, and CD8^+^ T cells was seen. Viral superantigen 7 (vSAG7)-reactive donor T cells formed clusters with host sialoadhesin^+^ liver macrophages [214]. Host liver macrophages played an essential role in the GvL effect [215].

Twenty-one days after treatment, neither live tumor tissue nor tumor infiltrating allogeneic T cells could be detected. Sites of previous metastases were replaced by scar tissue.

Three months after treatment, surviving mice still carried scar tissue from the primary tumor or they had rejected the tumor from the skin with wound healing underneath. This was followed by hair growth and reconstitution of normal fur [209].

This model system demonstrates complete remission of cancer in late-stage disease, including reversion of cachexia [6,211]. It should encourage future research in this direction.

## 10. Discussion

This story about mitochondria at work should be amazing not only for a scientist, but also for a practicing medical doctor. Mitochondrial dysregulations are often connected with chronic diseases, which are systemic. Such diseases require knowledge about mitochondria for diagnostic and therapeutic purposes, based on causal medicine.

Unfortunately, medical education provides only a superficial view on biochemistry, molecular biology, or immunology. The clinical education is organ oriented and prioritized by pharmacotherapy. In elderly patients with chronic diseases, polypragmasy (polypharmacotherapy) is often a result of multiorgan morbidity. It is clinically manifested by a reduction in the effectiveness of pharmacotherapy.

A change of paradigm seems necessary to improve the effectivity of the treatment of chronic diseases. This should include mitochondria-based medicine [11].

## 11. Summary

The reader is invited to watch small organelles at work in different types of cells of our body. These organelles, mitochondria, are semi-autonomous, and yet perfectly well integrated into the functioning of a eukaryotic cell. Most likely, they derived via endosymbiosis from bacteria of ancient times. Such bacteria were already equipped with a variety of enzymes and co-enzymes facilitating supply of energy and metabolites via the degradation of nutrients, such as carbohydrates and lipids by aerobic glycolysis, OXPHOS, and the electron transport chain. Mitochondria generate energy by providing ATP and are capable of metabolism. The dynamics of their metabolism includes cycles, such as the TCA and chains of redox enzymes in the inner mitochondrial membrane. Energy source serves hydrogen in form of its reduction equivalents, which are being burned via oxygen to generate H_2_O.

New insights into regulation are exemplified with cells from the muscle, liver, and brain. Other examples are immune cells, in particular T lymphocytes and stem cells from bone marrow. While mitochondria of heart muscle cells have optimized life-long ATP production to pump oxygen through our organs, mitochondria of liver cells focus on detoxification, including *OH radicals, and anabolic processes. Re-programming between energy supply and metabolism is essential for cells from the hematopoietic system such as HSCs and T cells. It involves signaling pathways such as PI3/Akt/mTOR, HIF-1α, MAPK, and NF-κB.

Dysregulation of mitochondrial function is an essential factor of chronic diseases: cardiovascular diseases (CVD), metabolic syndromes (MetS) (e.g., diabetes), neurodegenerative diseases (e.g., Parkinson, Alzheimer’s), and immune system diseases (e.g., autoimmunity, immune exhaustion, premature senescence, cytokine storm syndrome). Cancer is a major disease of dysregulation characterized by genetic and epigenetic mechanisms. These cause dysregulation of mitochondria leading to aerobic glycolysis and to a “truncated” TCA cycle. The review provides latest discoveries concerning mitochondria at work in the tumor microenvironment (TME) and during cancer cachexia. Examples of strategies for intervention are being presented with respect to CVD, MetS, immune system exhaustion, TME, and cachexia.

Figure 1 provides an overview of the interventional possibilities in various chronic diseases.

## 12. Conclusions

To learn from nature seems appropriate concerning the worldwide problems of energy supply and the environment. Billions of years of evolution have generated cellular organelles (chloroplasts), able to convert carbon (from CO_2_), molecular oxygen (O_2_), and molecular hydrogen (H_2_) with the help of photons and enzymes into glucose molecules. Other organelles (mitochondria) are able to degrade glucose with the help of O_2_, H_2_ and redox enzymes into ATP, H_2_O, and CO_2_. Eukaryotic cells most likely endocytosed bacteria with such biochemical capacities. Such symbiotic processes formed a basis for the establishment of multicellular organisms. It is a success story of co-evolution of plants and animals. Photosynthesis and cellular respiration would not have been possible without the help of bacteria equipped with redox enzymes to transfer electrons between different atoms. Perhaps, in the future, engineers might incorporate bacteria into their technologies to more effectively use hydrogen as energy source.

In recent years, scientists have contributed a lot to elucidate the function of mitochondria. Such knowledge is a prerequisite for understanding their dysregulation, especially in major chronic diseases: cardiovascular diseases, metabolic syndromes, neurodegenerative disorders, and dysfunctions within the hematopoietic and immune system. An important message is to support mitochondrial function as early as possible. Results from animal models suggest that even extreme dysregulatory situations, such as cancer cachexia, are not principally irreversible. It, thus, seems worth the effort to turn dysregulated mitochondria back into a regulatory physiological function.

## Figures and Tables

**Figure 1 biomedicines-08-00526-f001:**
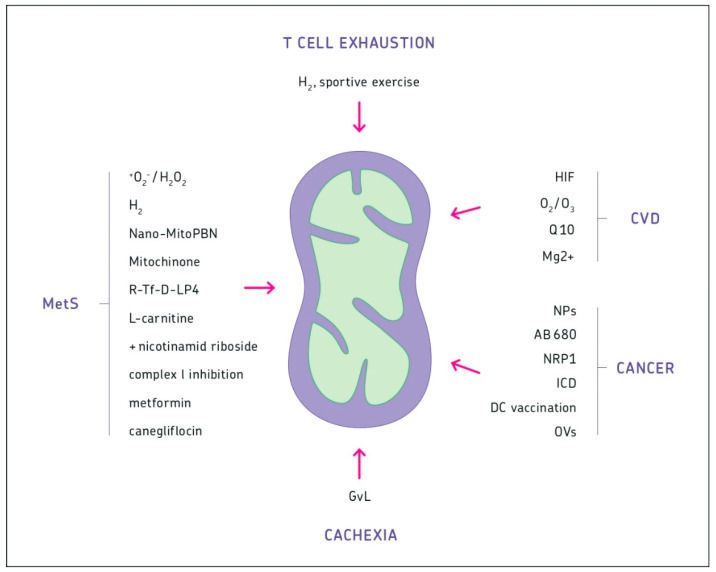
Mitochondria-targeted interventional strategies with metabolic syndromes (MetS), cardiovascular diseases (CVD), cancer (with tumor microenvironment), cancer cachexia, and T cell exhaustion. All interventional strategies are dealt with in the main text. H_2_ = molecular hydrogen; NPs = nanoparticles; ICD = Immunogenic cell death; DC vaccination = vaccination with dendritic cells expressing tumor antigens and eliciting immunological danger signals; GvL = Graft-versus-Leukemia effect exerted by immune T cells; OVs = oncolytic viruses.

**Table 1 biomedicines-08-00526-t001:** Energy production by bacteria.

Feature	Bacterium	Energy Source	Metabolism	Enzyme	Co-Enzyme	End Product
Glycolysis	Archaebacteria (e.g., Chromatiaceae)	H_2_S e^−^	Hexose to pyruvate	Ferredoxin-NADP-Reductase	Ferredoxin 2Fe-2S	S or S^−^ NADPH ATP
Glycolysis	Early Eubacteria	H_2_O e^−^	Hexose to pyruvate	GAPDH ^1^	NAD ^2^	NADH ATP
Fermentation	Early Eubacteria	H_2_O e^−^	Pyruvate to lactate	LDH ^3^	NAD	CO_2_ NADH ATP
Oxidative Phosphorylation, Cellular Respiration	Aerobic Eubacteria	H^+^ O_2_ e^−^	Pyruvate to TCA, Respiration chain	Cytochrome-c Oxidase	Cytochrome-c (Fe^3+^ to Fe^2+^)	CO_2_ H_2_O ATP
Photosynthesis	Cyanobacteria	H^+^ O_2_ Photons	Photosystem I Photosystem II	Plastocyanin: Ferredoxin-Oxidoreductase H_2_O: Plastoquinone Oxidoreductase	Chlorophyll P700 Chlorophyll P680	O_2_ NADPH ATP

^1^ GAPDH = glyceraldehyde-3-phosphate dehydrogenase; ^2^ NAD = nicotinamide adenine dinucleotide; ^3^ LDH = lactate dehydrogenase.

**Table 2 biomedicines-08-00526-t002:** Structural features of mitochondria and chloroplasts.

Feature	Mitochondrion	Chloroplast
Size and form	Like bacterium (2 mm)	Like bacterium (2 mm)
Inner membrane	Without 3-OH steroids	Without 3-OH steroids
Outer membrane	With 3-OH steroids With porin molecules Like eukaryotic cell	With 3-OH steroids With porin molecules Like eukaryotic cell
DNA	mtDNA ring without histone	ptDNA ring without histone
Replication	One start site	One start site
Copy number	About 10	About 100
RNA	rRNAs tRNAs mRNAs	rRNAs tRNAs mRNAs
Ribosomes	70S	70S
Start of protein	N-formyl-methionine	N-formyl-methionine
Inner membrane proteins Respiration chain	NADH dehydrogenase (p ^1^) Cytochrome c oxidase (p) ATP synthase (p)	Photosystem I (p) Photosystem II (p) ATP synthase (2 + 4)
Inner membrane lipid	Cardiolipin	Cardiolipin
Import from cell cytoplasm	Yes (proteins, sugar, fatty acids)	Yes (proteins, sugar, fatty acids)
Export to cell cytoplasm	No	No

^1^ p = partial; only a part of the enzyme polypeptide chains are encoded by mtDNA or ptDNA.

**Table 3 biomedicines-08-00526-t003:** Metabolic regulation of T lymphocytes.

Feature	Metabolic Program	Molecular Determinants
T cell co-stimulation	Diverse changes to augment mitochondrial mass and function	IL-15, CD28, PD1, CD137
T cell activation	Predominantly oxidative phosphorylation (OXPHOS)/aerobic glycolysis	Glut1 ^1^, lactic acid
Naive T cells Memory T cells	Mixed fuel oxidative phosphorylation Lipid oxidation	TRAF6 ^2^, AMPK ^3^
T cell subtypes		
Th1 Th2 Th9 Th17 Treg	Predominantly aerobic glycolysis/OXPHOS Lipid oxidation	mTORC1 ^4^ mTORC2 ^4^ Pu.1-TRAF6-NFkB, IL9 mir26 ^5^, HIF-1α ^6^, PKM2 ^7^, adiponectin Foxp3 ^8^, AMPK, BATF ^9^

^1^ Glut1 = glucose transporter 1; ^2^ TRAF6 = TNF receptor-associated factor 6; ^3^ AMPK = adenosine-monophosphate-activated protein kinase; ^4^ mTORC = mammalian target of rapamycin complex; ^5^ mir26 = distinct micro-ribonucleic acid (mirRNA); ^6^ HIF-1 = hypoxia inducible factor 1; ^7^ PKM2 = pyruvate kinase M2 isoform; ^8^ Foxp3 = forkhead box protein 3; ^9^ BATF = basic leucine zipper ATF-like transcription factor.

**Table 4 biomedicines-08-00526-t004:** New insights into cancer cachexia.

Feature	Site	Evidence	Reference
Mitochondrion Metabolism	Skeletal muscle Liver	Suppressed ACSL1 ^1^ OXPHOS proteome	[190,191]
Receptor signaling pathway	Skeletal muscle Skeletal muscle	SIRT1 ^2^-NOX4 ^3^ RAGE ^4^ and S008(L)	[192,193]
Transcription factor	Skeletal muscle	TWIST1 ^5^	[194]
Regulatory RNA	Skeletal muscle Adipose tissue Skeletal muscle	miRNA ncRNA miRNA-mRNA	[195,196,197,198]
Hormone	Hindbrain Adipose tissue	GDF15 ^6^ GFRAL-RET ^7^ receptor Asprosin, Leptin, Intellectin-1	[101,102,103,104,105,106,107,108,109,110,111,112,113,114,115,116,117,118,119,120,121,122,123,124,125,126,127,128,129,130,131,132,133,134,135,136,137,138,139,140,141,142,143,144,145,146,147,148,149,150,151,152,153,154,155,156,157,158,159,160,161,162,163,164,165,166,167,168,169,170,171,172,173,174,175,176,177,178,179,180,181,182,183,184,185,186,187,188,189,190,191,192,193,194,195,196,197,198,199,200,201,202,203,204]
Cytokine	Immune system	IL-6 ^8^	[205,206]
Immune cell Neutrophil Macrophage M2	Blood to brain Skeletal muscle	CCR2/CCL2 ^9^ CD163^+^	[207,208,209]

^1^ ACSL1 = acyl-CoA synthase long chain family member 1; ^2^ SIRT1 = silent information regulator 1; ^3^ NOX4 = nicotinamide adenine dinucleotide phosphate oxidase; ^4^ RAGE = recombinase-assisted genome engineering; ^5^ TWIST1 = TWIST family basic helix-loop-helix transcription factor 1; ^6^ GDF15 = growth differentiation factor 15; ^7^ GFRAL-RET = GDNF family receptor alpha like-RET proto-oncogene; ^8^ IL-6 = interleukin 6; ^9^ CCR2/CCL2 = a distinct chemokine receptor-chemokine ligand signaling pathway.

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
