# Peer review of "Mitochondria at Work: New Insights into Regulation and Dysregulation of Cellular Energy Supply and Metabolism"

_biomedicines, 2020, doi:10.3390/biomedicines8110526_

Round 1

Reviewer 1 Report

This submission is a rather broad and in-depth review of a mitochondrial energy function that includes a basic part of the comparison between bacteria and chloroplasts and pathological states in metabolic disease, neurodegenerative disease, and immunology and cancer. The review is clear and well written, and the structure is thoughtfully elaborated. This referee has only a few issues that should be addressed prior to final acceptance:

Major points:

  1. While this review comes already very complete, a chapter on the role of mitochondrial Ca2+ and its regulation for mitochondrial energy metabolism might complete this work
  2. While the review is very clear, its density is sometimes hard for readers, thus each chapter should be accompanied by summarizing graphics illustrating the adaptations, changes, and consequences.
  3. Please add a table on the new therapeutic strategies related to the related topics

Minor points:

  • There are some inconsistencies in the formatting that should be corrected
  • Tables are sometimes hard to be read and should be better aligned within the lines
  • Please use an entire title instead of “pathologic” or “physiologic”

Author Response

The revision contains a new figure, several new references and includes changes according

to the two reviewers. The positive comments of both reviewers encouraged further optimization

of the manuscript in order to cover the latest new insights into the many aspects of mitochondrial

function.

Response to Reviewer 1: 

Point 1 is delt with in lines 190-193 and lines 296-303.

Points 2+3: These points have initiated the construction of the new Figure.

All minor points were dealt with. The last minor point, see lines 276, 322 and 722.

Tables: The editing process has caused the problem with the tables. All tables have been re-

introduced as original versions. Table 1 shows both versions.

Response to Reviewer 2:

The first point: see lines 271-273.

The second point: see lines 421-428.

The third point: see lines 576-583, 984-992 and 1086-1089.

The spelling mistakes have been corrected.

Reviewer 2 Report

In the manuscript entitled “Mitochondria at work: New insights into regulation and dysregulation of cellular energy supply and metabolism”, Schirrmacher clearly reported the involvment of mitochondria in pathophysiological processes of major chronic diseases (cardiovascular diseases, metabolic syndrome, neurodegenerative diseases, immune system disorders and cancer) which are characterized by mitochondrial dysregulation.

The manuscript raises an interesting example of how in recent years scientists have contributed to elucidate the function of mitochondria and dimostrate that, for example, the extreme dysregulatory situations like cancer cachexia are not principally irreversible.

This review is interesting and well written. The article is well presented and the quality is adequated. The tables are well presented.

I'd ask the author to take into account the few points raised below:

  • Biological hydrogen production as an alternative energy source is an interesting paragraph of paper, however it is necessary to add may link to mitochondrial involvement.
  • Liver regeneration is a valuable point of review. Can be good to describe the mitochondrial involvment in a deep way.
  • Finally, to complete the picture of “mitochondria at work in our cells”, the author can briefly describe the role of transcription factors involved in cancer cell homeostasis and metabolic stress such as FoxO3A. Has been reported that in metabolically stressed cancer cells, FoxO3A is recruited to the mitochondria through activation of MEK/ERK and AMPK, where is imported and cleaved to reach mitochondrial DNA to activate expression of the mitochondrial genome to support mitochondrial metabolism. In cancer cells treated with chemotherapeutic agents, accumulation of FoxO3A into the mitochondria promoted survival in a MEK/ERK-dependent manner, while mitochondrial FoxO3A was required for apoptosis induction by metformin. Elucidation of FoxO3A mitochondrial vs. nuclear functions in cancer cell homeostasis might help devise novel therapeutic strategies to selectively disable FoxO3A prosurvival activity.
  • Line 271 “carciac” > cardiac
  • Line 285 “sarcomer” > sarcomere
  • Line 996 “NF B” > NFkB

Overall rating

The manuscript is recommended for publication with minor revision.

Author Response

The revision contains a new figure, several new references and includes changes according

to the two reviewers. The positive comments of both reviewers encouraged further optimization

of the manuscript in order to cover the latest new insights into the many aspects of mitochondrial function.

Response to Reviewer 1: 

Point 1 is delt with in lines 190-193 and lines 296-303.

Points 2+3: These points have initiated the construction of the new Figure.

All minor points were dealt with. The last minor point, see lines 276, 322 and 722.

Tables: The editing process has caused the problem with the tables. All tables have been re-introduced as original versions. Table 1 shows both versions.

Response to Reviewer 2:

The first point: see lines 271-273.

The second point: see lines 421-428.

The third point: see lines 576-583, 984-992 and 1086-1089.

The spelling mistakes have been corrected.